# Autocrine signaling by receptor tyrosine kinases in urothelial carcinoma of the bladder

**Young H. Lee[1], Molly M. Lee[1], Dinuka M. De Silva[1], Arpita Roy[1], Cara E. Wright[1], Tiffany K. Wong[1], Rene Costello[2], Oluwole Olaku[1], Robert L. Grubb, III[3], Piyush K. Agarwal[1], Andrea B. Apolo[2‡]\*, Donald P. Bottaro[1‡]\***

1 Urologic Oncology Branch, Center for Cancer Research, National Cancer Institute, National Institutes of Health, Bethesda, Maryland, United States of America, 2 Genitourinary Malignancies Branch, Center for Cancer Research, National Cancer Institute, National Institutes of Health, Bethesda, Maryland, United States of America, 3 Department of Urology, Medical University of South Carolina, Charleston, South Carolina, United States of America

‡ These authors are joint senior authors on this work.
\* don.bottaro@nih.gov (DPB); andrea.apolo@nih.gov (ABP)

**Data Availability Statement:** Expression array data have been deposited in the Gene Expression Omnibus (GEO) database of the National Center for Biotechnology Information, National Library of

## Abstract

Comprehensive characterizations of bladder cancer (BCa) have established molecular phenotype classes with distinct alterations and survival trends. Extending these studies within the tyrosine kinase (TK) family to identify disease drivers could improve our use of TK inhibitors to treat specific patient groups or individuals. We examined the expression distribution of TKs as a class (n = 89) in The Cancer Genome Atlas (TCGA) muscle invasive BCa data set (n >400). Patient profiles of potentially oncogenic alterations (overexpression and/or amplification) clustered TKs into 3 groups; alterations of group 1 and 3 TKs were associated with significantly worse patient survival relative to those without alterations. Many TK pathways induce epithelial-to-mesenchymal transition (EMT), which promotes tumor invasiveness and metastasis. Overexpression and/or amplification among 9 EMT transcriptional activators occurred in 43% of TCGA cases. Co-occurring alterations of TKs and EMT transcriptional activators involved most group 1 TKs; 24% of these events were associated with significantly worse patient survival. Co-occurring alterations of receptor TKs and their cognate ligands occurred in 16% of TCGA cases and several BCa-derived cell lines. Suppression of GAS6, MST1 or CSF1, or their respective receptors (AXL, MST1R and CSF1R), in BCa cell lines was associated with decreased receptor activation, cell migration, cell proliferation and anchorage independent cell growth. These studies reveal the patterns and prevalence of potentially oncogenic TK pathway-related alterations in BCa and identify specific alterations associated with reduced BCa patient survival. Detection of these features in BCa patients could better inform TK inhibitor use and improve clinical outcomes.

## Introduction

In 2018, 549,000 new cases of bladder cancer (BCa; urothelial (transitional) cell carcinoma of the bladder) and 200,000 bladder cancer-related deaths were estimated worldwide [1, 2].

Medicine, National Institutes of Health (URL: https://www.ncbi.nlm.nih.gov/geo/), Series GSE156348. Original uncropped and unadjusted images underlying blot or gel results reported in this submission have been deposited in the Zenodo public data repository: https://doi.org/10.5281/zenodo.4749396 All other relevant data are within the manuscript and its S1 File.

**Funding:** This work was supported by the Intramural Research Program of the Center for Cancer Research, National Cancer Institute, National Institutes of Health (URL https://ccr.cancer.gov/), through grant awards ZIA-BC011095-12 and ZIA-BC011124-12 to DPB. The funders had no role in study design, data collection and analysis, decision to publish, or preparation of the manuscript.

**Competing interests:** D. Bottaro is an inventor on US Government held patents that are generally related to the subjects reported here: US 10,035,833: "Vascular Endothelial Growth Factor Antagonists and Methods of Making"; US 9,550,818 and related international publications WO/2013/163606: "Methods for Use of Vascular Endothelial Growth Factor Antagonists"; US 8,617,831: "Methods for Diagnosing and Monitoring the Progression of Cancer by Measuring Soluble c-Met Ectodomain"; US 7,964,365 and related international WO/2007/056523: "Methods for Diagnosing and Monitoring the Progression of Cancer"; US 8,304,199: Methods for Diagnosing and Monitoring the Progression of Cancer by Measuring Soluble c-Met Ectodomain"; US 8,569,360 and related international publications WO/2009/124024 and WO/2009/124013: "Compositions and Methods for Inhibition of Hepatocyte Growth Factor Receptor c-Met Signaling": US 7,871,981 and related international WO/2001/028577: "Inhibition of Cell Motility, Angiogenesis and Metastasis". This does not alter our adherence to PLOS ONE policies on sharing data and materials. No other authors have competing interests to disclose.

**Abbreviations:** TK, tyrosine kinase; RTK, receptor TK; ctRTK, cabozantinib-targeted RTK; TKI, TK inhibitor; BCa, bladder cancer; MIBC, muscle-invasive bladder cancer; OS, overall survival; PFS, progression-free survival; TCGA, The Cancer Genome Atlas; EMT, epithelial-to-mesenchymal transition; TA, transcriptional activator.

Although 70% of newly diagnosed disease is confined to the mucosa, recurrence and progression are frequent, and long-term surveillance is required. The remaining 30% of new cases are more advanced, involving muscle invasion, lymph node involvement or distant metastases (mUC). Standard of care combination platinum-based chemotherapy for mUC patients provides a median overall survival (OS) of 9–15 months [3, 4]. Several immune checkpoint inhibitors have been approved by the U.S. Food and Drug Administration (FDA) in the past 3 years for platinum-refractory patients [5–8], 2 of which are also approved as first-line therapy for cisplatin-ineligible mUC patients with high levels of programmed death-ligand 1 protein [9, 10]. Despite some durable responses, the overall response rate to these therapies is 14–23% [5–9]. More recent FDA approvals of erdafitinib for platinum refractory mUC patients with gene alterations of *FGFR3* or *FGFR2*, enfortumab vedotin, and sacituzumab govitecan for platinum- and immune checkpoint inhibitor-refractory mUC patients [11–13] represent continued progress, but more effective identification and targeting of pathways that drive oncogenesis in this disease is urgently needed.

Several comprehensive molecular interrogations of BCa patient tumor samples have improved our understanding of disease pathogenesis, revealed functionally relevant molecular phenotypes with parallels to those of other cancers, and provided a foundation for more detailed analyses of specific signaling pathways as likely oncogenic drivers among these phenotypes and in individuals [14–22]. With the goal of identifying targets for molecular diagnosis and treatment where diagnostic reagents and targeted drugs already exist, we found that positive results from a phase II NCI clinical trial of the multikinase inhibitor cabozantinib for patients with advanced BCa (NCT01688999) implicated several members of the tyrosine kinase (TK) superfamily in disease progression [23]. We surveyed 12 BCa-derived cell lines for evidence of oncogenic signaling by a perceived primary cabozantinib target, MET, the receptor TK (RTK) for hepatocyte growth factor (HGF) [24]. No *MET* gene alterations or copy number variations among these cell lines are recorded in COSMIC [25], none of the cell lines produced HGF [24], and significant co-overexpression of *MET* and *HGF* transcripts occurred only twice in the recent TCGA data (408 cases) analyzed by Robertson et al. [22]. Together these findings suggested that ligand-independent or autocrine MET signaling occurs infrequently in BCa, and that cabozantinib targets other than MET might be driving oncogenesis. This theory was reinforced by the lack of any significant association between tumor tissue MET content or kinase activation and outcome in trial NCT01688999 [23], where patients with heavily pre-treated, platinum-refractory metastatic urothelial carcinoma with measurable disease and bone metastases displayed clear clinical responses to single-agent cabozantinib treatment [23]. Cabozantinib also exerted innate and adaptive immunomodulatory activity in patients and in supporting cell-based studies [23], providing a strong rationale for combining cabozantinib with immunotherapeutic agents, and for identifying active cabozantinib targets in BCa. A phase 1 study led by our group reported significant efficacy for this combination in multiple genitourinary tumors including bladder and kidney cancer [26], leading to a phase 3 of this combination that resulted in FDA approval of cabozantinib and nivolumab for the first-line treatment of clear cell renal cell carcinoma [27].

The reported cabozantinib targets include 15 RTKs encoded by *AXL*, *CSF1R*, *FLT1*, *FLT3*, *FLT4*, *KDR*, *KIT*, *MET*, *MERTK*, *MST1R*, *NTRK1*, *NTRK2*, *RET*, *ROS1* and *TEK* [28–36]. We report here results of kinase profiling of cabozantinib *in vitro* that implicate the 4 RTKs encoded by *DDR1*, *DDR2*, *NTRK3* and *TYRO3* as additional high affinity targets. We found that potentially oncogenic gene alterations (amplification, overexpression and/or mutation) of these 19 cabozantinib-targeted RTKs (ctRTKs) occur in 66% of 408 cases in the TCGA BCa database analyzed by Robertson et al. [22], as determined using tools available on the cBioPortal [37]. RNASeq and gene amplification data in this set for *AXL, CSF1R, DDR2, KDR, MST1R,*

*PDGFRA* and *TEK* show significant co-occurrence of >2-fold expression and/or gene amplification for 8 cognate ligands of these receptors in a combined 16% of cases, suggestive of oncogenic autocrine RTK activation. Results obtained using BCa-derived cell lines indicate that autocrine signaling via the GAS6/AXL, MST1/MST1R, or CSF1/CSF1R pathways drives cell migration and proliferation, effects that were suppressed by ligand- or RTK-specific RNAi and blocked by cabozantinib. These findings reveal the frequency and patterns of autocrine RTK signaling in BCa and suggest that detection of these events in BCa patients could better inform TK inhibitor (TKI) use and thereby improve clinical outcomes.

## Materials and methods

### Reagents and cell lines

Tissue culture media and supplements were obtained from Invitrogen (Carlsbad, California USA). Human bladder cancer-derived cell lines were obtained from American Type Culture Collection (ATCC; Manassas, VA USA): 5637 (ATCC HTB-9), J82 (ATCC HTB-1), HT1376 (ATCC CRL-1472), UMUC3 (ATCC CRL-1749), T24 (ATCC HTB-4), TCCSUP (ATCC HTB-5), RT4 (ATCC HTB-2) and SW780 (ATCC CRL-2169), or from the European Collection of Authenticated Cell Cultures (available through Millipore Sigma-Aldrich, St. Louis, MO USA): RT112 (85061106), UMUC5 (UM-UC-5, 08090502) and HT1197 (87032403). The T24-derived cell lines T24T, T24M2, SLT3 and FL3 were provided by Dan Theodorescu and originally described by Nicholson et al. [38]. The cell line MDXC1 was derived from an explant of a J82 xenograft tumor grown in a SCID/BEIGE mouse for 50 days. The cell lines MDXC2, MDXC31R, MDXC31L, MDXC33R, MDXC33L, MDXC34R, and MDXC34L are explant derivatives of J82 tumors serially passaged twice (MDXC2) or three times (all others) grown in SCID/BEIGE mice for 50 days/passage. Lines were confirmed to be 100% human in origin by RT-PCR analysis with multiple probes for human and mouse genes. Lines were confirmed to be mycoplasma-free using the MycoSEQ Mycoplasma Detection System (cat. #s 4452222 and 4460626) from Thermo Fisher Scientific (Waltham, Massachusetts USA) using the manufacturer's protocol. Antibodies against phospho-Met (1234/1235; cat. #3126), pErk (cat. #4370), tErk (cat. #4695), pAkt (cat. #4060), tAkt (cat. #4685), AXL (cat. #8661), MST1R (cat. #2654), DDR1 (cat. #3917), DDR2 (cat. #12133), PTK7 (cat. #25618), MERTK (cat. #4319) and GAPDH (cat. #5174) were obtained from Cell Signaling Technology (Danvers, Massachusetts USA). Antibodies against Met (cat. # BAF358, AF276), RYK (cat. #AF4907), CSF1 (cat. #AF216), CSF1R (cat. #MAB3291) and GAS6 (cat. # AF885) were obtained from R&D Systems (Minneapolis, Minnesota USA). Cabozantinib and TP0903 were obtained from the NCI Repository of Chemical Agents—Small Molecules and Isolated Natural Products of the Developmental Therapeutics Program, Division of Cancer Treatment and Diagnosis, National Cancer Institute, Bethesda, Maryland USA (https://dtp.cancer.gov/repositories/).

### TCGA database analyses

Several prior studies describe potentially oncogenic gene mutations in BCa [14–22]. TK overexpression, in contrast, has received far less scrutiny and was our primary focus here, particularly in the context of co-occurring alterations in other pathway components that support the possibility of oncogenic pathway activation. mRNA expression (RNA-Seq V2 RSEM) data of tumor samples from muscle-invasive bladder cancer (MIBC) patients described in the dataset analyzed by Robertson et al. [22], was downloaded from cBioPortal website [37] (https://www.cbioportal.org/study/summary?id=blca_tcga_pub_2017; April, 2019) using links therein with the following option settings: "Bladder Cancer (TCGA, Cell 2017)" was chosen from https://www.cbioportal.org/; "Query by Gene" was selected; "Select Genomic Profile" was set to

"mRNA Expression, z-score +/- 2"; "Select Patient/Case Set" was set to "All Samples"; in the "Genes of Interest" box, the 89 TKs listed in S1A Table in S1 File were entered using HUGO gene identifiers. The complete downloaded RNA-Seq data set is presented in S1B Table in S1 File. The same dataset was also analyzed using tools available on the cBioPortal website for gene mutations, copy number alteration (GISTIC 2.0) and mRNA expression level (absolute z-score > 2-fold; RNA Seq V2 RSEM) for selected TKs, cognate ligands, and transcription factors. Co-occurrence of gene amplification and/or overexpression was evaluated by Fisher's exact test with significance at $p < 0.05$ and $q < 0.05$. Overall and progression-free survival for cases with and without alterations was assessed by the Kaplan-Meier method and compared by log-rank test; in all cases, case sets defined by altered and unaltered gene sets were compared across the entire patient dataset (n = 408). RNA Seq data was imported into Qlucore Omics Explorer software (versions 3.0–3.5, Qlucore AB, Lund, Sweden) to identify differential gene expression patterns with selected p and q values, perform 2-group and multi-group statistical tests, and produce heat maps with clustering by specific annotations (e.g., molecular phenotype classifications defined previously [22]) as noted in the text.

## Gene silencing and RT-PCR

All siRNA used for in vitro studies were synthesized by GE Dharmacon (Lafayette, Colorado USA). siRNA transfections used Lipofectamine RNAiMax according to the manufacturer's protocol (Thermo Fisher Scientific, Waltham, Massachusetts USA). For quantitative RT-PCR measurements, total RNA was obtained using the RNeasy kit (Qiagen, Valencia, California USA) and concentrations were determined spectroscopically at 260 nm using a NanoDrop ND-1000 (Thermo Fisher Scientific). Real-time quantitative PCR was performed using a QuantStudio 6 Flex real-time PCR system (Applied Biosystems, Foster City, California USA) following manufacturer's protocols. All PCR primer and siRNA sequences are listed in S6 Table in S1 File. Relative gene expression levels were evaluated using the delta-delta CT method. Absolute mRNA copy number was determined for selected targets in BCa cell lines by using purified mRNA for RT-qPCR reactions and including parallel samples for PCR amplification of the neomycin gene in serial dilutions of pcDNA 3.1 plasmid. Neomycin gene PCR results were used to generate a reference standard curve relating PCR product concentration to cycle number and comparing the last cycle number at exponential product generation (Cq) for all other reactions to that of the reference standard. Selected PCR reactions were analyzed on 1.5% agarose gels in Tris-EDTA buffer and bands were visualized using ethidium bromide.

## SDS-PAGE, immunoblot analysis and 2-site immunoassays

For SDS-PAGE and immunoblot analysis, cultured cells were washed with cold PBS, extracted in Laemmli sample buffer, sonicated in ice-cold water with 3 cycles of 15 sec sonication followed by 1 min of cooling time. Samples were then heated for 5 min at 95˚C prior to SDS-PAGE and electrophoretic transfer to nitrocellulose or PVDF membrane. Membranes were processed as described previously [24]: membranes were blocked with 5% milk in Tris buffered saline, 0.1% Tween 20 (TBST) for 1 h at 25˚C before incubating with primary antibody (1:1000 dilution) in TBST-5% milk for 16 h at 4˚C. Membranes were then washed 3 times with TBST, incubated with horse radish peroxidase-labeled secondary antibody (1:10,000 dilution) for 1 h at 25˚C, and washed for 3 h with TBS prior to ECL detection (Pierce/Thermo Fisher Scientific) according to the manufacturer's supplied protocol. Imaging and quantitation of ECL light emission was performed using an Azure Biosystems c600 imaging system (Dublin, California USA).

Electrochemiluminescent 2-site immunoassays for Met and Axl total protein and phospho-protein content in cell extracts were performed as described previously [39]. Briefly, streptavidin-coated 96-well plates manufactured for use in a Meso Scale Discovery (MSD) S600 Sector Imager (Meso Scale Discovery, Gaithersburg, Maryland USA) were first coated with 300 microliters/well I-Block solution (Applied BioSystems Thermo Fisher Scientific cat. # A1300), washed 3 times with PBS (150 microliters/well), then incubated with biotin-tagged, affinity purified capture antibody (diluted in 0.5% BSA in PBS; 5 micrograms/ml, 25 microliters/well) for 1 h with shaking. Wells were washed 3 times with PBS before adding samples or standards. Samples were prepared from intact cells at 80% confluence that were washed twice with cold PBS before lysis in cold 100 mM HEPES buffer (pH 7.4) containing 1% Triton X-100 and protease and phosphatase inhibitors. Cell extracts were clarified by high-speed centrifugation at 4°C and 25 microliter aliquots were added to wells of 96-well plates at a total protein concentration of 500 micrograms/ml for 1 h at 25°C with shaking. Recombinant protein reference standards (100 microliters/well) for Met (R&D Systems cat. # 358MT) or Axl (R&D Systems cat. # 154AL) were added in parallel to generate curves from 0.01 ng/ml to 100 ng/ml in semi-log increments. Wells were washed 3 times with PBS before adding detection antibody labeled with MSD Sulfotag (diluted in 0.5% BSA in PBS) at 1 microgram/ml, 25 microliters/well for 1 h with shaking. Wells were then washed 4 times with PBS before adding MSD Read Buffer T with surfactant (150 microliters/well) and then read immediately in an MSD S600 Sector Imager. All samples were measured in quadruplicate. Mean values from control wells lacking specific capture antibody were subtracted from all other raw values and standard curves were constructed by plotting sample signal intensity against recombinant protein standard concentration. A non-linear regression curve-fitting algorithm (GraphPad Prism software versions 6–8) was used to generate an equation from which sample values for Met or Axl concentration were derived from background-corrected mean signal intensity values. Mean values among groups were compared for statistically significant differences using unpaired t-test (paired human cell lines) or analysis of variance; $R^2$ and/or P values are provided in the text and/or figure legends. Electrochemiluminescent immunoassays for Akt, Erk and phosphorylated forms of these proteins were performed similarly in 96-well format using kits available from Meso Scale Discovery (Gaithersburg, Maryland USA).

## Cell migration, cell proliferation and anchorage-independent growth assays

Migration assays were performed using 8 micron pore size Corning Transwell inserts (Corning, New York USA) according to the manufacturer's instructions. Images were captured by light microscopy and image analysis and quantitation were performed using Image J software V1.47 (National Institutes of Health, Bethesda, Maryland USA). For proliferation assays, cells were transfected with siRNA negative control, siAXL, siGas6, siMST1R or siMST1 and plated in triplicate at a density of $2.5 \times 10^4$ cells/35 mm dish in defined medium. Gas6, MST1, and/or cabozantinib were added on days 1, 2, and 4. Cells were detached and counted using a hemocytometer on day 3 or 6. Anchorage independent growth was measured as described previously [39] with the following modifications: A base layer of 0.5% Noble agarose (Difco, Franklin Lakes, New Jersey USA) in phenol red-free DMEM was added to 96 well plates. Cells in 0.3% agarose in phenol red-free DMEM were added on top of the base layer. Cells were fed with DMEM with or without growth factors (as noted in the text) and cabozantinib every other day. After 1 week, MTT was added to quantify the viable colonies by absorbance using a PerkinElmer Victor plate reader (PerkinElmer, Hopkinton, Massachusetts USA). Significant differences between 2 groups were determined by Student's t test using GraphPad Prism software versions 6–8, where p < 0.05 was considered statistically significant.

## Tumorigenicity studies in mice

All experiments involving animals were performed in accordance with NIH Guidelines for Care and Use of Laboratory Animals and conforming to ARRIVE guidelines using institutionally reviewed and approved protocol UOB-009 (D. Bottaro, Principal Investigator) at the Center for Cancer Research, National Cancer Institute, Bethesda, Maryland, United States of America. Animal health was monitored daily for pain, distress or generalized discomfort and classified categorically as (1) minimal, transient or no pain or distress, or (2) pain or distress relieved by appropriate measures (analgesics e.g., buprenorphine); or (3) having unrelieved pain or generalized discomfort. Humane endpoints: any animals in category (3), including animals appearing moribund, cachectic, exhibiting ulcerous skin lesions or unable to obtain food or water, were euthanized immediately using an in-house line of carbon dioxide gas supplied to a closed chamber. For the animal studies described here, J82 and J82-derived cell lines were injected subcutaneously ($10^6$ cells in 100 microliters) into SCID/BEIGE mice (Charles River Laboratories, Wilmington, Massachusetts USA; n = 10 per group) and tumor volumes were measured at regular intervals using calipers (3 dimensions) as described previously [24]. Study endpoint: animals were euthanized when tumors reached 2 cm in any single dimension or a calculated volume of 500 $mm^3$, or at 50 days post-implantation. Tumors were removed for cell culture and implantation into new mice using sterile surgical procedures performed in a laminar flow hood. Animals implanted with tumor fragments received perioperative anesthesia (isoflurane, 0.5–4%, via inhalation), perioperative care, aseptic surgical subcutaneous implantation, and post-operative care accordingly to the ACUC approved protocol. A licensed veterinarian was present in the animal facility or on-call continuously throughout the study; no mortality occurred outside of the planned euthanasia or humane endpoints of this study. Tumor growth curves were fitted by regression analysis using GraphPad Prism software versions 6–8.

## mRNA expression, bioinformatic and intracellular pathway analyses

The Nanostring PanCancer Progression 770 gene expression panel (Nanostring Technologies Inc, Seattle, Washington USA) was used to analyze J82 cells and the J82 tumor xenograft-derived cell line MDXC1. Cells were grown to near confluence, serum-deprived for 16 h, total RNA was extracted, and samples were processed and analyzed per the manufacturer's protocol. Hybridized panels (770 genes and 30 control genes) were read at maximum field count (555 FOV). Data was normalized using Nanostring NSolver software versions 2.0 or 3.0 with a background threshold set to 20 counts, normalization reference set to housekeeping (control) genes, and normalization factor set to geometric mean. NSolver normalized expression data from J82 and MDXC1 cell lines was imported into Qlucore Omics Explorer software V3.0–3.6 for 2-group comparisons with J82 samples set to control and filtering for q < 0.05 and p < 0.0021, generating a list of 321 significantly differentially expressed genes and heatmap with hierarchical clustering. Expression array data have been deposited in the Gene Expression Omnibus (GEO) database of the National Center for Biotechnology Information, National Library of Medicine, National Institutes of Health (URL: https://www.ncbi.nlm.nih.gov/geo/), Series GSE156348.

For Ingenuity Pathway Analysis (https://www.qiagenbioinformatics.com /products/ingenuitypathway-analysis, Qiagen NV, Venlo, Netherlands), the Nanostring 770 gene data file was uploaded, and pre-processing statistical cutoffs were set to q < 0.05 and p < 0.0021, enabling IPA to identify the same 321 gene list produced by Qlucore as "analysis ready". The list was then processed using IPA Core Analysis (version 49932394, Nov 2019) with the following settings: *Reference set* = user data file (770 genes); *Relationship to include*: Direct and

Indirect; Does not Include Endogenous Chemicals; *Data Sources* = All; *Species* = All; *Tissues and Cell Lines* = All; *Mutation* = All; *Filter Summary*: Consider only relationships where confidence = Experimentally Observed; no cutoffs were set for fold change or ratio. P values for overlap between gene expression changes and IPA Molecule Groups, Functions, Activities or Pathways were derived using the right-tailed Fisher's exact test with Benjamini and Hochberg multiple test correction where appropriate. A positive IPA z-score indicates direct concordance between the direction (expression increase or decrease) and of genes altered in the sample and genes included in the IPA subcategory, a negative score indicates an inverse concordance. The IPA z-score is unrelated to the magnitude of expression change, provided it exceeds a threshold based on p-value and q-value (false discovery rate) [40].

## Results

### TK gene expression in BCa TCGA samples and BCa cell lines

Data from muscle-invasive bladder cancer (MIBC) tumor samples in the bladder urothelial carcinoma TCGA dataset analyzed by Robertson et al. [22] were analyzed using cBioPortal tools [37] and downloaded for further analyses. Samples with RNA Seq V2 data (n = 408) were queried for potentially oncogenic alterations (excluding mutations) in 89 TKs (>98% of the human protein tyrosine kinome, S1 Table in S1 File), which occurred at combined frequencies of 53% for gene amplification, 97% for mRNA overexpression with z-score >2, and 55% for overexpression with z-score >4 (S2A Table in S1 File). Collectively these alterations were not associated with significant survival differences at log-rank p<0.05, but trends of lower OS and disease-free survival (DFS) for the altered groups were noted (S2A Table in S1 File).

The mRNA expression profiles of 52 TKs varied significantly among the 5 molecular phenotypes of BCa developed previously [22]: neuronal (N), basal squamous (BS), luminal (L), luminal infiltrating (LI), and luminal papillary (LP), forming 2 distinct patterns (Fig 1A). Thirty-one TKs (*ABL1/2*, *AXL*, *BTK*, *CSF1R*, *DDR2*, *EPHA3*, *EPHB2/3/4*, *FER*, *FGFR1*, *FLT3*, *FYN*, *HCK*, *IGF1R*, *ITK*, *JAK1/2/3*, *LCK*, *LYN*, *MET*, *PDGFRA/B*, *PTK7*, *ROR1/2*, *TEK*, *TIE1* and *ZAP70*, hereafter referred to as group 1) were highly expressed in the worse OS phenotypes N and BS, with lower expression in LP, the best OS phenotype, whereas 21 TKs (*DDR1*, *EPHA1*, *EPHB6*, *ERBB2/3/4*, *FGFR2/3*, *INSR*, *LMTK2*, *MERTK*, *MST1R*, *PTK2/6*, *SRC*, *SRMS*, *STYK1*, *TNK1/2*, *TXK*, and *TYK2*, hereafter group 2) showed a reciprocal expression pattern among those phenotypes (p = $1.00 \times 10^{-4}$, q = $1.01 \times 10^{-4}$, $F_{2,405} \geq 9.42$, $R^2 \geq 0.044$ for the 3-group comparison F test: LP *vs.* (LI+L) *vs.* (BS+N); Fig 1). The expression profiles of the remaining 37 TKs varied independently of molecular phenotype (hereafter group 3; S1 Table in S1 File).

The frequency of potentially oncogenic alterations (excluding mutations) in these 3 TK groups was similar: significant mRNA overexpression (z-score >2) and/or gene amplification of group 1 TKs occurred in 67% of samples, among group 2 TKs in 69% of samples and among group 3 TKs in 74% of samples (S3A–S3C Table in S1 File). However, significant (q<0.05) co-occurrence of these alterations was disproportionately higher among group 1 TKs: there were 422 mRNA overexpression and/or gene amplification co-occurrences in 63 of 465 (14%) of group 1 TK pairings (S3A Table in S1 File), *vs.* 114 co-occurrences in 12 of 210 (6%) of group 2 TK pairings (S3B Table in S1 File), and 115 co-occurrences in 16 of 666 (2.4%) of group 3 TK pairings (S3C Table in S1 File). Median OS and DFS for patients harboring these alterations relative to those without alteration also differed among the TK groups, and were consistent with reported OS differences [22] between molecular phenotypes. Patients with overexpression (z-score >2) and/or gene amplification of group 1 TKs (most frequent in N and BS phenotypes) had significantly worse median OS (28.22 vs. 61.40 mos., log-rank p = 0.0308) and DFS (27.99 vs. 82.42 mos., log-rank p = 0.0145; S2B Table in S1 File, Fig 2A)

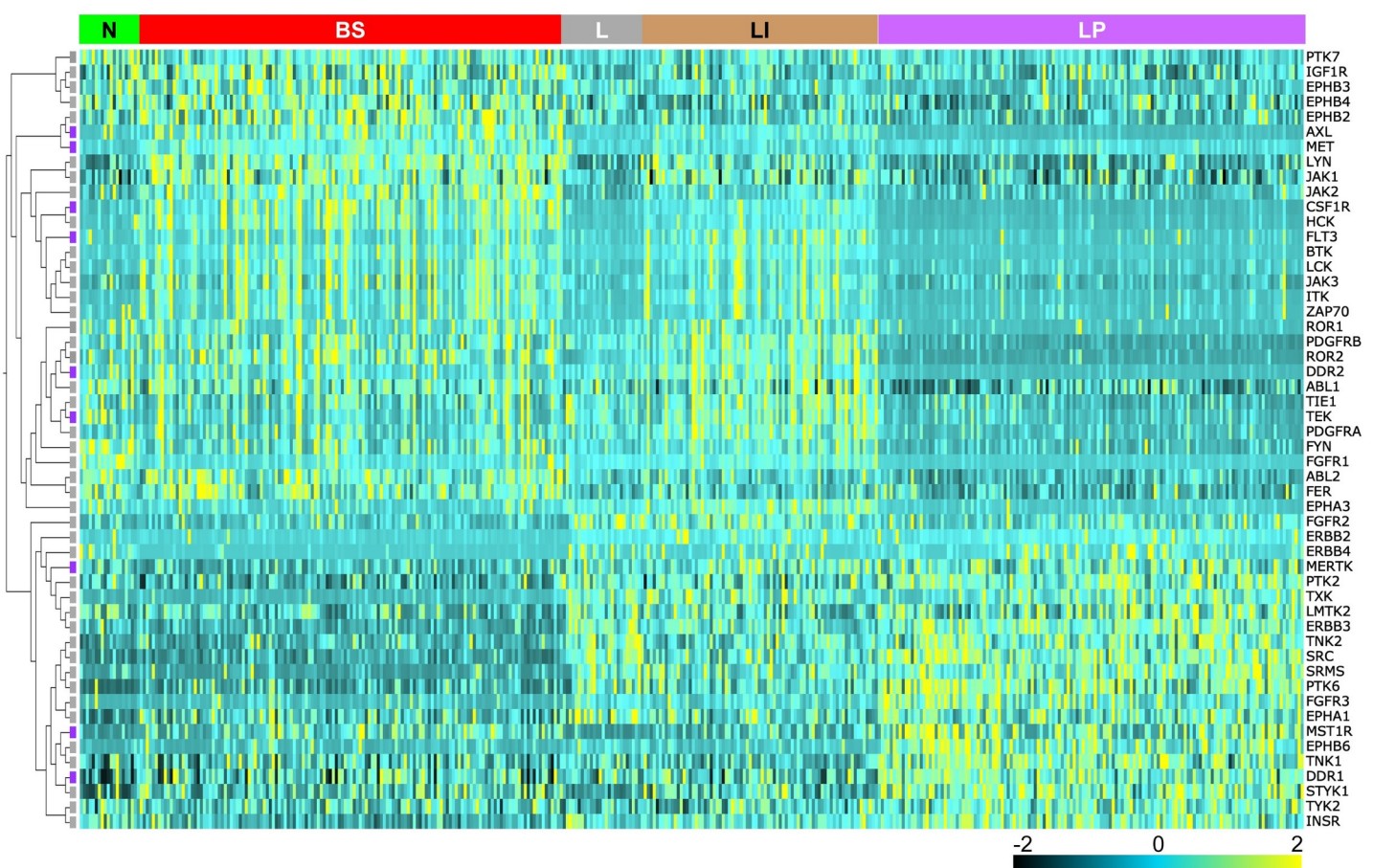

**Fig 1. Heat map of relative TK mRNA expression levels among MIBC patient tumor samples.** TCGA data for MIBC patient tumor samples (n = 408) reported previously [22]. TK gene symbols are listed vertically at right. Samples are grouped on the horizontal axis by mRNA subtype classifications as defined by Robertson et al. [22] indicated at the top (from left to right): neuronal (N, green); basal squamous (BS, red); luminal (L, gray); luminal infiltrating (LI, gold); luminal papillary (LP, purple). TK genes are clustered hierarchically; in the cladogram at left cabozantinib targets are designated with violet squares.

than those without alteration, whereas patients with these alterations in group 2 TKs (most frequent in L, LI and LP phenotypes) had significantly better median OS (46.65 vs. 22.14 mos., log-rank p = 3.197e-4) and DFS (51.41 vs. 19.05 mos., log-rank p = 6.673e-4) than those without alteration (S2C Table in S1 File, Fig 2B). Patients with overexpression (z-score >2) and/or amplification of group 3 TKs, which varied independent of molecular phenotype, showed a trend of worse median OS (30.91 vs. 59.26 mos., log-rank p = 0.134) and significantly worse median DFS (27.99 vs. 72.34 mos., log-rank p = 0.0391) than those without alteration (S2D Table in S1 File, Fig 2C).

The TK expression profiles of 15 BCa-derived cell lines were compared to those of TCGA tumor samples using mRNA copy number for 31 TKs from groups 1 and 2 (Fig 3A). These 31 TKs (19 from group 1 and 12 from group 2, preserving the 3:2 ration of the larger 52 gene set) were selected for practicality as the minimum number that retained the same significant correlation with molecular phenotype as the parent 52 TK set (p = 1.00 x10$^{-4}$, q = 8.97 x 10$^{-5}$, $F_{2,405} \geq$ 9.42, $R^2 \geq$ 0.044 for the 3-group comparison F test: LP *vs.* (LI + L) *vs.* (BS +N); Fig 3A). Each cell line profile was correlated with each TCGA sample profile, correlation coefficients were averaged within each molecular phenotype, and cell lines were assigned to N, BS, LI, L or LP phenotype by best average coefficient that was distinguished from coefficients for other

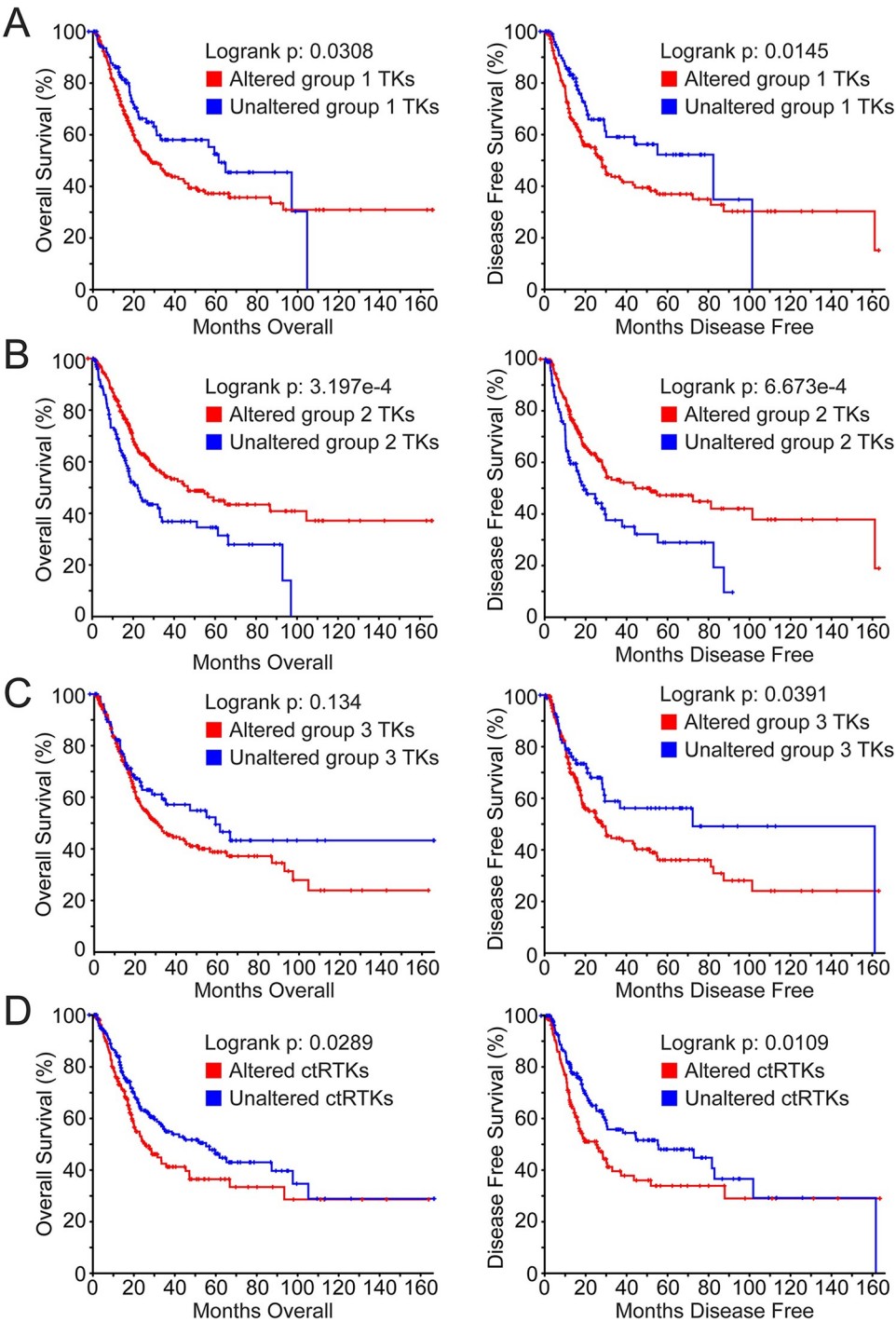

**Fig 2. Kaplan-Meier survival analyses in patients harboring significant mRNA overexpression and/or amplification of TK genes.** Kaplan-Meier analyses of OS (left panels) and DFS (right panels) in patients harboring significant mRNA overexpression (z-score >2) and/or amplification of **(A)** group 1 TKs (red) *vs.* those without alteration (blue; median OS 28.22 *vs.* 61.40 mos., log-rank p = 0.0308; median DFS 27.99 *vs.* 82.42 mos., log-rank p = 0.0145); **(B)** group 2 TKs (red) *vs.* those without alteration (blue; 46.65 *vs.* 22.14 mos., log-rank p = 3.197e-4; median DFS 51.41 *vs.* 19.05 mos., log-rank p = 6.673e-4); **(C)** group 3 TKs (red) *vs.* those without alteration (blue; 30.91 *vs.* 59.26 mos., log-rank p = 0.134; median DFS 27.99 *vs.* 72.34 mos., log-rank p = 0.0391); **(D)** 14 ctRTKs (AXL, CSF1R, FLT3, KDR, MET, MERTK, NTRK1/2/3, RET, ROS1, TEK, DDR2 and TYRO3; red) *vs.* those without alteration (blue; 25.56 *vs.* 54.86 mos., log-rank p = 0.0289; median DFS 25.23 *vs.* 55.16 mos., log-rank p = 0.0109).

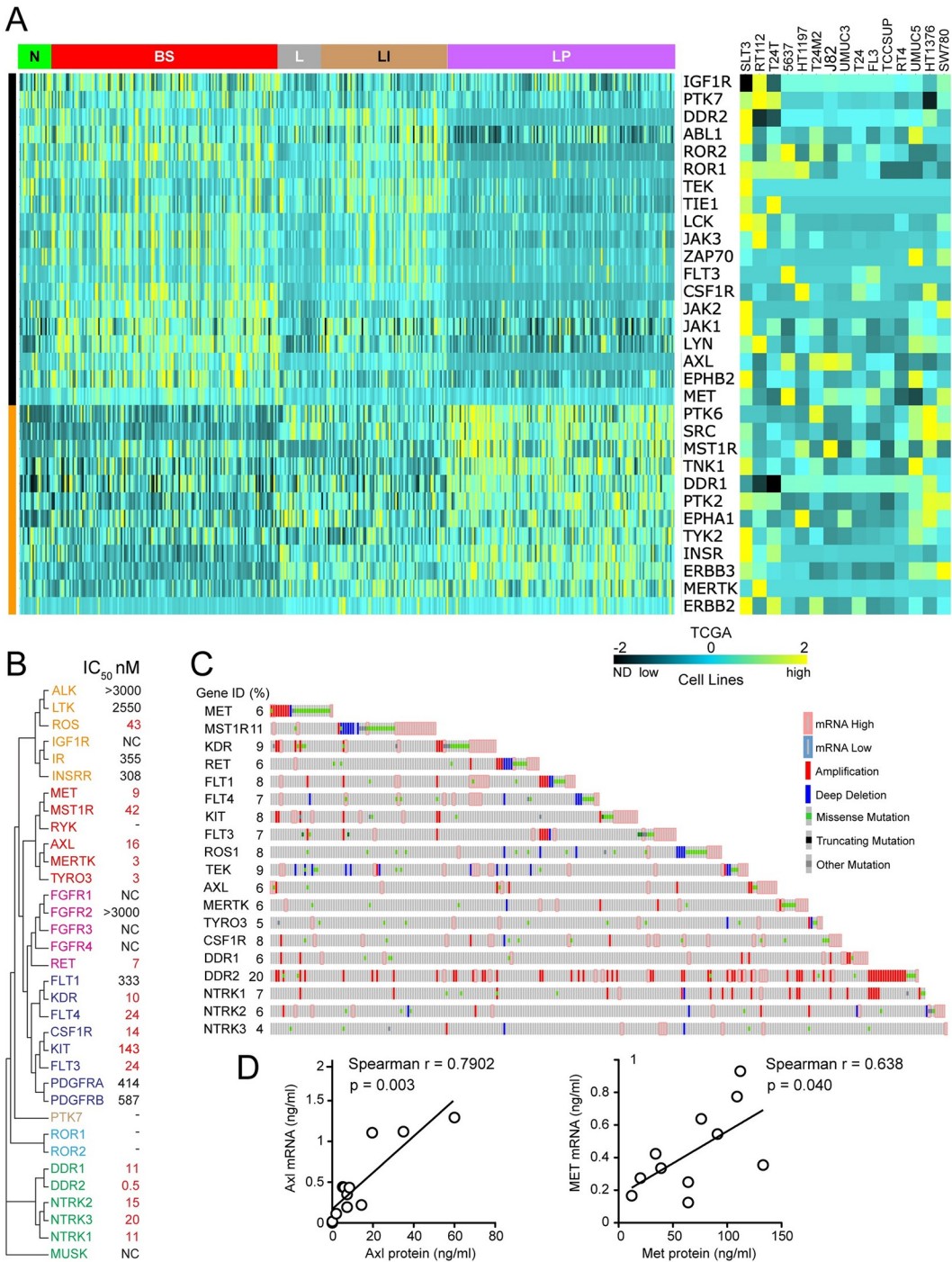

**Fig 3. TK expression in MIBC patient tumor samples and BCa cell lines. (A)** Heat maps of relative mRNA expression levels in 408 tumor samples from patients with MIBC [22] (RNA Seq V2, left) and absolute mRNA copy number in 15 BCa cell lines (right) of 31 TKs (gene symbols listed vertically, middle). MIBC patient tumor samples are grouped on the horizontal axis by mRNA subtype classifications as previously defined [22] indicated at the top (from left to right): neuronal (N, green); basal squamous (BS, red); luminal (L, gray); luminal infiltrating (LI, gold); luminal papillary (LP, purple). TK genes are clustered hierarchically; colored vertical bars at indicate group 1 TKs (black) above group 2 TKs (orange). Highest expression is shown in yellow, moderate expression in blue and lowest expression in black; ND for cell line data indicates transcript was undetectable. **(B)** Cladogram of 34 receptor TKs (RTKs) most closely related to Met based on TK domain amino acid sequence identity, constructed using the EMBL-EBI ClustalW2 Phylogeny tool (http://www.ebi.ac.uk/Tools/phylogeny/clustalw2_phylogeny). RTK symbol colors identify established subfamilies. Values for cabozantinib doses calculated to provide 50% TK inhibition (IC$_{50}$, nM) in a cell-free assay, listed at right, were derived from a minimum of 3 x

10-dose binding curves with regression (R) values > 0.98 unless indicated by NC (not calculated), where values were >3 micromolar. Values considered clinically relevant (< 150 nM) are shown in red. TKs for which IC$_{50}$ values could not be determined are denoted by a single dash. **(C)** Oncoprint showing alterations for 19 genes encoding ctRTKs (identified previously and/or by cell-free kinase inhibition assay) among 408 MIBC samples [22]. Gene symbols and percentage of samples altered (%) for each are listed at left (Oncoprint generated using visualization tools at the cBioPortal for Cancer Genomics, Memorial Sloan Kettering Cancer Center, https://www.cbioportal.org/ [37]). **(D)** Linear regression analysis of *AXL* (left) and *MET* (right) mRNA abundance (y-axis) *vs*. protein content (x-axis) for 15 BCa-derived cell lines shows significant direct correlation (*AXL*: Spearman r = 0.638, p = 0.040; *MET*: Spearman r = 0.7902, p = 0.003).

phenotypes by t-test (Table 1). By these criteria, 10 of the 15 cell lines were assigned to a single phenotype or 2 related phenotypes, whereas the remaining 5 failed the t-test threshold but most closely resembled a single phenotype (Table 1). The same classification was generated using the related method of hierarchically clustering cell TK mRNA profiles by heat map (Fig 3A, right).

Positive interim results from our phase II clinical trial of the multikinase inhibitor cabozantinib for patients with advanced BCa (NCT01688999) [23], prompted us to analyze TKs targeted by cabozantinib in the TCGA MIBC dataset [22]. The 16 reported cabozantinib targets (encoded by *AXL, CSF1R, FLT1, FLT3, FLT4, KDR, KIT, MET, MERTK, MST1R, NTRK1, NTRK2, RET, ROS1, TEK* and *TYRO3*) had been identified in a variety of ways [28–36]. We compared targets on a single assay platform using cell-free cabozantinib dose-dependence studies to obtain IC$_{50}$ values for 30 RTKs encompassing known targets (except TEK) and members of structurally related RTK subfamilies (Fig 3B). In this context, cabozantinib inhibited most of the 16 known target RTKs with IC$_{50}$ values below 150 nM, and 3 others (DDR1, DDR2, and NTRK3) that were not previously reported (Fig 3B), bringing the total to 19 targets. IC$_{50}$ values could not be determined for RYK, ROR1/2 and PTK7, so whether they are inhibited by cabozantinib remains to be determined.

**Table 1. Classification of BCa-derived cell lines based on TK-specific molecular phenotypes adapted from those defined using TCGA data analyzed by Robertson et al. [22] (all samples with RNASeq V2 data, n = 408).**

| Cell line | Closest phenotype | Correlation value | Passed t-test |
|-----------|-------------------|-------------------|---------------|
| RT4 | Luminal papillary | 0.152 | Yes |
| UMUC5 | Luminal papillary | 0.112 | Yes except Luminal |
| SW780 | Luminal papillary | 0.247 | Yes except Luminal |
| HT1376 | Luminal papillary | 0.298 | Yes except Luminal |
| FL3 | Basal squamous | 0.140 | Yes |
| UMUC3 | Basal squamous | 0.176 | Yes |
| TCCSUP | Basal squamous | 0.134 | No |
| J82 | Basal squamous | 0.101 | Yes |
| T24 | Basal squamous | 0.049 | Yes |
| T24M2 | Luminal infiltrating | 0.035 | No |
| 5637 | Basal squamous | 0.110 | Yes |
| HT1197 | Basal squamous | 0.042 | Yes |
| T24T | Neuronal | 0.197 | No |
| RT112 | Neuronal | 0.107 | No |
| SLT3 | Neuronal | 0.058 | No |

Expression profiles for 31 TKs among TCGA patient samples were significantly correlated with molecular phenotype (neuronal, basal squamous, luminal, luminal infiltrating, and luminal papillary) and OS (p = 1.00 x10-4, q = 8.97 x 10−5, F$_{2,405}$ > 9.42, R$^2$ > 0.044 for the 3-group comparison F test: LP vs. (LI + L) vs. (BS +N). Cell line expression profiles for the TKs were compared with the respective TCGA patient tumor TK profiles and assigned to a molecular phenotype if its mean correlation coefficient was distinguished from the respective coefficients for the other phenotypes by t-test at p < 0.05.

**Table 2. Significant co-occurrence of potentially oncogenic alterations (RNASeq V2 >2-fold and/or gene amplification) for RTKs and cognate ligands in BCa TCGA dataset analyzed by Robertson et al. (n = 408) [22] and differences in median OS and PFS.**

| Gene | | | Amplification and/or Overexpression | | | | Odds Ratio (Log 2) | p-Value |
|---|---|---|---|---|---|---|---|---|
| TK Group | RTK | Ligand | Neither | RTK | Ligand | Both | | |
| 1 | AXL | GAS6 | 358 | 16 | 23 | 7 | 2.768 | <0.001 |
| 1 | CSF1R | CSF1 | 355 | 21 | 17 | 11 | >3 | <0.001 |
| 1 | DDR2 | COL4A1 | 310 | 62 | 21 | 11 | 1.389 | 0.016 |
| 1 | DDR2 | COL10A1 | 309 | 70 | 15 | 10 | 1.557 | 0.013 |
| 3 | KDR | VEGFA | 340 | 30 | 27 | 7 | 1.555 | 0.026 |
| 2 | MST1R | MST1 | 364 | 17 | 14 | 9 | >3 | <0.001 |
| 1 | PDGFRA | PDGFA | 347 | 30 | 21 | 6 | 1.725 | 0.024 |
| 1 | TEK | ANGPT2 | 364 | 22 | 13 | 5 | 2.670 | 0.004 |

All 19 ctRTKs displayed potentially oncogenic alterations (overexpression, amplification and mutations unclassified as to pathogenic consequence) harbored by 282/408 (69%) of cases in the MIBC TCGA dataset [22] and ranging in frequency from 4% (*NTRK2/3*, *TYRO3*) to 20% (*DDR2*); homozygous deletions were rare by comparison, affecting 13 of these genes in 51/408 (12.5%) of cases (Fig 3C). The mRNA expression distributions for 9 of 19 ctRTKs were significantly different among BCa molecular phenotypes; of these, 6 were group 1 TKs (*AXL*, *CSF1R*, *DDR2*, *FLT3*, *MET* and *TEK*) and 3 were group 2 TKs (*DDR1*, *MERTK* and *MST1R*; Fig 1A); the remaining 10 ctRTKs (*FLT1/4*, *KDR*, *KIT*, *NTRK1/2/3*, *RET*, *ROS1* and *TYRO3*) were in TK group 3. Significantly co-occurring potentially oncogenic alterations (overexpression, amplification and mutations unclassified as to pathogenic consequence, 150 events) involved 15/171 (9%) of ctRTK pairings; these events involved only TK groups 1 (67% of events) and 3 (33% of events; S3D Table in S1 File, top). All 15 BCa-derived cell lines examined above expressed 1 or more ctRTK (Fig 3A, right). Patients harboring overexpression and/or amplification among 14 ctRTKs (46% of cases) had significantly worse OS (25.56 *vs.* 54.86 mos., log-rank p = 0.0289) and DFS (25.23 *vs.* 55.16 mos., log-rank p = 0.0109) than those without alteration (S2E Table in S1 File, Fig 2D). This subset of 14 ctRTKs excludes the group 2 members DDR1, MERTK and MST1R, as group 2 TK overexpression was associated with improved survival relative to groups 1 and 3. Significantly co-occurring alterations in the 14 ctRTKs (60 events in 6.6% of pairings) involved *AXL*, *CSF1R*, *DDR2*, *MET*, *TEK*, *KDR* and *NTRK1* (S3D Table in S1 File, bottom). Several ctRTKs showed general concordance between mRNA content (Fig 3A, right) and protein abundance detected by immunoblotting or 2-site immunoassay (Fig 3D).

### Potential autocrine RTK signaling pathways in BCa tumors and cell lines

cBioPortal tools were used to identify significant co-occurrence of mRNA overexpression (>2-fold RNASeqV2 z-scores) and/or gene amplification for specific RTK/cognate ligand pairs in 65/408 cases (16%) in the TCGA dataset [22] involving 7 RTKs (*AXL*, *CSF1R*, *DDR2*, *KDR*, *MST1R*, *PDGFRA* and *TEK*), suggestive of oncogenic autocrine RTK signaling (Table 2). For these RTKs and *NTRK2*, heat map analysis also revealed broad concordance between receptor and cognate ligand expression levels among BCa TCGA cases, including 5 ligands of *DDR2* (*COL1A1*, *COL3A1*, *COL4A1*, *COL5A1* and *COL10A1*) (Fig 4A, left). Quantitative mRNA analyses for these ligand transcripts revealed a similar pattern of co-expression of *AXL/ GAS6*, *CSF1R/CSF1*, *DDR2/COL-1A1*, *-3A1*, *-4A1*, *5A1*, *-10A1*, *MST1R/MST1*, and *NTRK2/ NTF4* in a combined 20% of 15 BCa-derived cell lines (Fig 4A, right). Receptor/ligand co-expression events were confirmed by immunoblot analysis for CSF1R/CSF1, AXL/GAS6, and MST1R/MST1 in 9 BCa cell lines (Fig 4B); 7 of these cell lines most closely resembled the

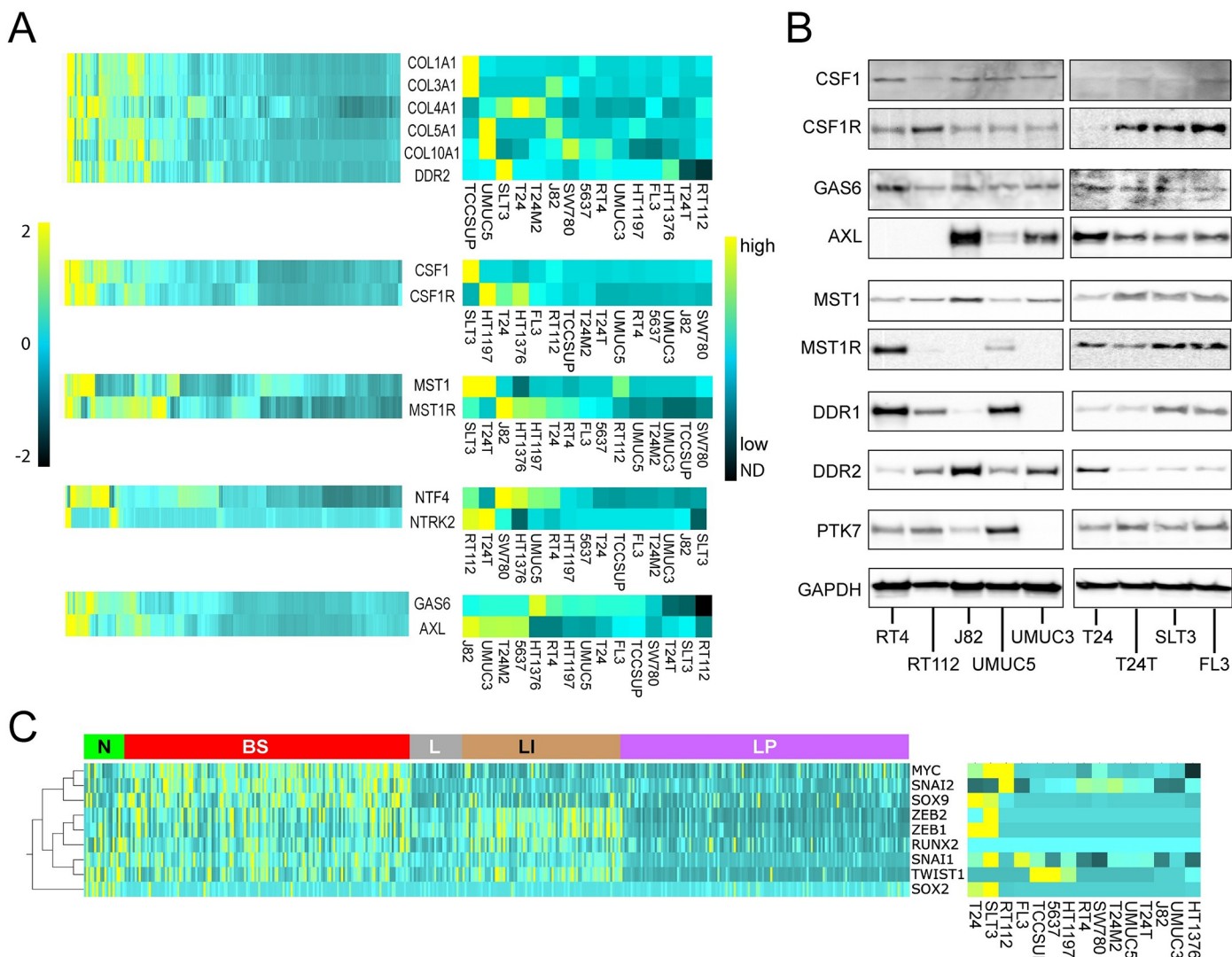

**Fig 4. Potential autocrine RTK signaling in MIBC patient samples and BCa cell lines. (A)** Heat map showing relative mRNA expression levels of ctRTK-cognate ligand pairs (RNA Seq V2) among 408 tumor samples from patients with MIBC [22] (left), and among 15 BCa-derived cell lines (absolute mRNA copy number, right). **(B)** Immunoblot detection of ctRTKs and their cognate ligands, CSF1/CSF1R, GAS6/AXL, MST1/MST1R, and evidence of DDR1, DDR2 and PTK7 proteins in 9 BCa-derived cell lines. GAPDH (bottom panel) serves as a loading control. **(C)** Left: heat map showing relative mRNA expression levels of 9 EMT-associated transcriptional activators (listed at right) among 408 tumor samples from patients with MIBC [22]. Samples are grouped on the horizontal axis by mRNA subtype classifications defined previously [22] indicated at the top (from left to right): neuronal (N, green); basal squamous (BS, red); luminal (L, gray); luminal infiltrating (LI, gold); luminal papillary (LP, purple), and clustered hierarchically on the vertical axis. Right: heat map showing absolute mRNA copy levels of 9 EMT-associated transcriptional activators among 15 BCa-derived cell lines (identified at bottom).

molecular phenotypes associated with lowest OS (BS and N, Table 1) [21], and the CSF1R/ CSF1, AXL/GAS6, and MST1R/MST1 pathways showed the highest Odds Ratio (Log 2) values for co-occurrence (Table 2). Consistent with TCGA patient data, DDR1, DDR2 and PTK7 were also detected frequently among the 9 BCa cell lines (Fig 4B). The group 1 TK DDR2 has been implicated recently in immune evasion by BCa tumors [41], whereas targeting PTK7 (also a group 1 TK) has been reported to induce regression of several tumor model systems [42]. Five of 7 RTKs involved in significant ligand co-occurrence are in group 1, *MST1R* is in group 2 and *KDR* in group 3, and 6 of 7 RTKs (excepting *PDGFRA*) are targeted by cabozantinib. Most RTK/ligand pairs were associated with lower median OS and DFS values relative to

**Table 3. Median months OS and DFS for combined significantly concurrent potentially oncogenic alterations (RNASeq V2 >2-fold and/or gene amplification and/or mutation as noted) for RTKs and cognate ligands in BCa TCGA dataset (n = 408) [22].**

| RTK/Ligand: Analysis Conditions | OS or DFS | Group | Cases, Total | Cases, Deceased or Recurred Progressed | Median Months | Logrank p |
|---|---|---|---|---|---|---|
| AXL/CSF1R/PDGFR + Ligands: exp>2, amp | OS | Co-altered | 23 | 13 | 23.62 | 0.2320 |
| | | Unaltered | 382 | 164 | 34.95 | |
| | DFS | Co-altered | 19 | 11 | 18.00 | 0.1610 |
| | | Unaltered | 300 | 130 | 32.59 | |
| AXL/CSF1R/PDGFR + Ligands: exp>2, amp, mut | OS | Co-altered | 25 | 14 | 23.62 | 0.2180 |
| | | Unaltered | 380 | 163 | 34.95 | |
| | DFS | Co-altered | 21 | 13 | 18.00 | 0.0606 |
| | | Unaltered | 298 | 128 | 32.59 | |
| All RTK/Ligands w/out MST1R/MST1: exp>2, amp | OS | Co-altered | 36 | 18 | 24.28 | 0.2320 |
| | | Unaltered | 369 | 159 | 35.38 | |
| | DFS | Co-altered | 29 | 16 | 18.00 | 0.1090 |
| | | Unaltered | 290 | 125 | 32.59 | |
| All RTK/Ligands w/out MST1R/MST1: exp>2, amp, mut | OS | Co-altered | 46 | 20 | 27.04 | 0.7600 |
| | | Unaltered | 359 | 157 | 34.03 | |
| | DFS | Co-altered | 38 | 24 | 18.00 | 0.0101 |
| | | Unaltered | 281 | 117 | 43.96 | |

*exp>2: mRNA overexpression z-score > 2-fold; amp: gene amplification; mut: missense mutation, unknown significance; NA: not available.

unaltered cases, but did not meet a 5% significance threshold by log-rank test (S2F Table in S1 File, center column). The notable exception was the group 2 MST1R/MST1 pair, for which Kaplan-Meier analysis displayed a trend of improved survival relative to unaltered cases (S2F Table in S1 File, center column). Combined, cases harboring groups 1 and 3 RTK/ligand co-alterations had worse OS and DFS relative to unaltered cases with log-rank p values approaching 5% but exceeded this threshold only for DFS (18.00 *vs.* 43.96 mos., p = 0.0101) for expression z >2.0 and/or gene amplification and/or RTK gene mutation of unclassified pathogenic consequence (Table 3).

Many TK pathways induce epithelial-to-mesenchymal transition (EMT), which is thought to be a critical process in promoting tumor invasiveness and metastasis. Additional indirect evidence of TK pathway activation and oncogenic signaling in BCa was obtained by interrogating the TCGA dataset [22] for coincident overexpression of TKs and EMT transcriptional activators (TAs) and by measuring TA mRNA copy number in the 15 BCa cell lines identified above. Heat map analysis of TCGA and cell line data revealed a pattern of higher expression for 9 EMT TAs (*MYC, RUNX2, SNAI1, SNAI2, SOX2, SOX9, TWIST1, ZEB1* and *ZEB2)* among molecular phenotypes with lower survival, consistent with oncogenic impact (Fig 4C). Potentially oncogenic alterations (>2-fold mRNA overexpression and/or gene amplification) in these EMT TAs occurred at a combined rate of 174/408 TCGA samples (43%) and significant co-occurrence rate of 14% (Table 4). Median OS for those 14% harboring EMT TA co-occurrence was significantly lower than for unaltered cases (27.43 vs. 44.91 mos., p = 0.0175).

**Table 4. Significant co-occurrence of potentially oncogenic alteration (overexpression per RNASeq V2 >2-fold, and/or gene amplification) among EMT TAs in TCGA dataset analyzed by Robertson et al. (n = 408) [22] and associated differences in median OS and PFS.**

| Gene | | Potentially Oncogenic Alteration | | | | Odds Ratio (Log 2) | p value | Significant difference in OS and/or PFS (altered vs unaltered) |
|---|---|---|---|---|---|---|---|---|
| EMT TA 1 | EMT TA 2 | Unaltered | TA 1 | TA 2 | Both | | | |
| ZEB1 | ZEB2 | 363 | 16 | 13 | 12 | >3 | <0.001 | PFS 15.54 *vs*. 36.86, p = 0.0303 |
| MYC | SNAI2 | 337 | 35 | 17 | 15 | >3 | <0.001 | None |
| TWIST1 | TWIST2 | 371 | 29 | 1 | 3 | >3 | 0.002 | OS 19.68 *vs*. 35.38, p = 0.0250 |
| | | | | | | | | PFS 16.39 *vs*. 37.78, p = 2.491e-5 |
| RUNX2 | ZEB1 | 366 | 10 | 23 | 5 | 2.992 | 0.002 | OS 19.38 *vs*. 38.21, p = 5.605e-3 |
| SNAI1 | TWIST1 | 357 | 15 | 26 | 6 | 2.457 | 0.004 | PFS 17.35 *vs*. 37.78, p = 9.118e-3 |
| TWIST1 | ZEB2 | 353 | 26 | 19 | 6 | 2.1 | 0.009 | PFS 16.39 *vs*. 37.78, p = 7.601e-4 |
| RUNX2 | ZEB2 | 368 | 11 | 21 | 4 | 2.672 | 0.010 | OS 18.65 *vs*. 41.72, p = 0.0320 |
| | | | | | | | | PFS 14.29 *vs*. 36.86, p = 0.0145 |
| SNAI1 | TWIST2 | 381 | 19 | 2 | 2 | >3 | 0.015 | None |
| TWIST2 | ZEB2 | 377 | 2 | 23 | 2 | >3 | 0.020 | PFS 10.35 *vs*. 32.59, p = 5.323e-3 |
| SOX9 | TWIST1 | 346 | 26 | 26 | 6 | 1.619 | 0.031 | PFS 17.97 *vs*. 43.96, p = 4.762e-3 |

Potentially oncogenic alterations co-occurred significantly for 22 (71%) group 1 TKs and 7 EMT TAs; 9 of these 38 concurrent alterations (24%) were associated with significantly lower OS and/or progression-free survival (PFS) relative to unaltered cases (Table 5A). In contrast, potentially oncogenic alterations co-occurred among only 5 (24%) group 2 TKs and 5 EMT TAs, none of which were associated with diminished survival (Table 5B).

## Autocrine AXL, CSF1R and MST1R signaling in BCa cell lines

Potential autocrine signaling by *AXL*, *CSF1R* and *MST1R* pathways in BCa cell lines was further investigated using exogenous ligand to induce activation, using cabozantinib to block RTK activation, and using short interfering RNAs (siRNAs) to suppress expression of *GAS6* in J82 cells, and *MST1* and *CSF1* in FL3 cells (Fig 5). Axl kinase activation (observed as autophosphorylation) normalized to total Axl protein (pAxl/tAxl), was stimulated >2-fold by exogenous Gas6 added to intact serum-deprived J82 cells, and this was suppressed significantly below control levels by added cabozantinib (Fig 5A, left). Consistent with autocrine signaling by endogenous Gas6 production (Fig 4B, middle), J82 cells transfected with siGAS6 also showed significant pAxl/tAxl suppression relative to control siRNA transfected cells (Fig 5A, right). The same pattern of significant exogenous Gas6-induced activation, suppression below baseline by cabozantinib, and siGAS6-associated baseline suppression, was observed for the downstream effectors Akt (Fig 5B) and Erk (Fig 5C). These results indicate receptor proximal pathway functionality and support the presence of autocrine signaling. This approach was also used to interrogate the receptor proximal pathways for MST1R and CSF1R in FL3 cells (Fig 5D–5F). These pathways were activated by exogenously added ligand and suppressed significantly below baseline by siRNAs directed against their cognate ligands or by cabozantinib, supporting the presence of autocrine proximal pathway activation in these cells.

Gas6-, MST1-, and CSF1-induced migration of J82 and FL3 cells, respectively, was also repressed by cabozantinib treatment (Fig 6A and 6B). siRNA suppression of *AXL* (Fig 6A, inset) significantly reduced J82 cell migration below the level of untreated serum-deprived cells (Fig 6A), indicating that autocrine Gas6/Axl signaling contributed to the migration of otherwise untreated J82 cells. In FL3 cells, comparable manipulation of the MST1 and CSF1 pathways using exogenous ligand, cabozantinib and RTK-directed siRNA also indicated autocrine driven migration in otherwise untreated cells (Fig 6B). Exogenously added Gas6

**Table 5. (A) Significant co-occurrence of potentially oncogenic alteration (overexpression >2-fold, and/or gene amplification) for Group 1 TKs and EMT TAs in TCGA dataset (n = 408) [22] and associated median OS and PFS.** (B) Significant co-occurrence of potentially oncogenic alteration (overexpression >2-fold, and/or gene amplification) for Group 2 TKs and EMT TAs in TCGA dataset (n = 408) [22] and associated median OS and PFS.

**A**

| Gene | | Potentially oncogenic alteration | | | | Odds Ratio (Log 2) | P value | Significant difference in OS and/or PFS (altered vs unaltered) |
|---|---|---|---|---|---|---|---|---|
| EMT TA | Group 1 TK | Not altered | EMT TA | TK | Both | | | |
| MYC | MET | 336 | 43 | 16 | 9 | 2.136 | 0.002 | None |
| | LYN | 333 | 42 | 19 | 10 | 2.061 | 0.002 | None |
| RUNX2 | AXL | 366 | 15 | 19 | 4 | 2.361 | 0.018 | OS 18.89 *vs.* 41.72, p = 0.0112 |
| | | | | | | | | PFS 15.5 *vs.* 36.86, p = 0.0172 |
| | FGFR1 | 342 | 13 | 43 | 6 | 1.876 | 0.018 | None |
| | FYN | 365 | 15 | 20 | 4 | 2.283 | 0.02 | None |
| SNAI1 | TIE1 | 346 | 17 | 33 | 8 | 2.303 | 0.002 | None |
| | FGFR1 | 333 | 22 | 35 | 14 | 2.598 | <0.001 | None |
| | ABL2 | 337 | 29 | 31 | 7 | 1.392 | 0.04 | None |
| | LYN | 350 | 25 | 18 | 11 | >3 | <0.001 | None |
| SOX2 | EPHB3 | 351 | 12 | 25 | 16 | >3 | <0.001 | None |
| | FYN | 350 | 30 | 19 | 5 | 1.618 | 0.046 | None |
| | PTK7 | 357 | 31 | 12 | 4 | 1.941 | 0.041 | None |
| TWIST1 | ABL1 | 353 | 24 | 19 | 8 | 2.631 | <0.001 | PFS 17.35 *vs.* 36.86, p = 5.563e-3 |
| | DDR2 | 305 | 19 | 67 | 13 | 1.639 | 0.004 | PFS 19.09 *vs.* 43.17, p = 0.0335 |
| | PDGFRA | 347 | 21 | 25 | 11 | 2.862 | <0.001 | PFS 17.35 *vs.* 37.78, p = 2.209e-3 |
| | PDGFRB | 353 | 25 | 19 | 7 | 2.379 | 0.002 | OS 19.68 *vs.* 34.95, p = 0.0424 |
| | | | | | | | | PFS 15.34 *vs.* 37.78, p = 8.717e-5 |
| | TIE1 | 339 | 24 | 33 | 8 | 1.776 | 0.009 | OS 26.91 *vs.* 41.72, p = 0.0256 |
| | | | | | | | | PFS 16.79 *vs.* 37.78, p = 1.361e-3 |
| | TEK | 351 | 26 | 21 | 6 | 1.948 | 0.014 | PFS 16.79 *vs.* 36.86, p = 4.532e-4 |
| | ROR2 | 350 | 26 | 22 | 6 | 1.876 | 0.016 | None |
| TWIST2 | FYN | 379 | 1 | 21 | 3 | >3 | <0.001 | None |
| | ABL1 | 375 | 2 | 25 | 2 | >3 | 0.024 | None |
| | PDGFRA | 366 | 2 | 34 | 2 | >3 | 0.041 | PFS 18.00 *vs.* 36.86, p = 0.0312 |
| | PDGFRB | 376 | 2 | 24 | 2 | >3 | 0.022 | PFS 10.05 *vs.* 36.86, p = 6.630e-4 |
| | TEK | 375 | 2 | 25 | 2 | >3 | 0.024 | None |
| ZEB1 | ABL1 | 348 | 29 | 15 | 12 | >3 | <0.001 | None |
| | DDR2 | 301 | 23 | 62 | 18 | 1.926 | <0.001 | None |
| | CSFR1 | 341 | 31 | 22 | 10 | 2.322 | <0.001 | None |
| | AXL | 347 | 34 | 16 | 7 | 2.159 | 0.005 | None |
| | FYN | 346 | 34 | 17 | 7 | 2.067 | 0.006 | None |
| | ITK | 344 | 35 | 19 | 6 | 1.634 | 0.031 | None |
| | FGFR1 | 323 | 32 | 40 | 9 | 1.183 | 0.044 | None |
| | ROR1 | 348 | 30 | 15 | 11 | >3 | <0.001 | None |
| | TEK | 347 | 30 | 16 | 11 | 2.991 | <0.001 | None |
| | TIE1 | 335 | 28 | 28 | 13 | 2.474 | <0.001 | None |
| | PDGFRA | 339 | 29 | 24 | 12 | 2.547 | <0.001 | None |
| | PTK7 | 354 | 34 | 9 | 7 | >3 | <0.001 | None |
| | PDGFRB | 345 | 33 | 18 | 8 | 2.216 | 0.002 | None |
| | ROR2 | 343 | 33 | 20 | 8 | 2.056 | 0.004 | None |

*(Continued)*

**Table 5.** (Continued)

**B**

| Gene | | Potentially oncogenic alteration | | | | Odds Ratio (Log 2) | P value | Significant difference in OS and/or PFS (altered vs. not) |
|---|---|---|---|---|---|---|---|---|
| EMT TA | Group 2 TK | Not altered | EMT TA | TK | Both | | | |
| MYC | PTK2 | 333 | 33 | 19 | 19 | >3 | <0.001 | None |
| | TXK | 340 | 44 | 12 | 8 | 2.365 | 0.002 | None |
| SNAI1 | PTK2 | 341 | 25 | 27 | 11 | 2.474 | <0.001 | None |
| | TYK2 | 356 | 20 | 23 | 5 | 1.952 | 0.022 | None |
| SNAI2 | PTK2 | 341 | 25 | 27 | 11 | 2.474 | <0.001 | None |
| SOX2 | TNK2 | 352 | 14 | 24 | 14 | >3 | <0.001 | None |
| | TYK2 | 353 | 23 | 23 | 5 | 1.738 | 0.035 | None |
| ZEB1 | DDR1 | 344 | 35 | 19 | 6 | 1.634 | 0.031 | None |

significantly increased 5637 and J82 cell proliferation rates, and cabozantinib treatment significantly inhibited proliferation by these lines below the rate of untreated cells, in the presence or absence of added Gas6 (Fig 6C). In FL3 cells, added MST1 alone significantly increased proliferation rate, and cabozantinib treatment alone significantly reduced proliferation rate, relative to untreated cells (Fig 6D). But unlike Gas6-driven proliferation in 5637 and J82 cells, treatment with both cabozantinib and MST1 reduced FL3 proliferation to the control cell growth rate, but not below (Fig 6D). Autocrine-driven proliferation by the *AXL*, *MST1R* and *CSF1R* pathways in J82 and FL3 cells was further supported by studies in which siRNA suppression of ligands or RTKs (Fig 7A) was exerted in the presence or absence of exogenous ligand and/or cabozantinib (Fig 7B–7D). Consistent with the results of biochemical and cell migration studies, J82 cells transfected with siRNA directed against *GAS6* or *AXL* showed significantly reduced proliferation relative to control siRNA transfected cells (Fig 7B), as did FL3 cells transfected with siRNA directed against *MST1*, *MST1R*, *CSF1*, or *CSF1R* (Fig 7C and 7D). Anchorage independent growth by J82 (Fig 7E) and FL3 (Fig 7F) was similarly enhanced significantly by added ligand, inhibited by cabozantinib treatment, and suppressed by RTK-directed siRNA.

## Phenotypic changes in J82 cells upon serial xenograft passage in mice

Mouse xenograft studies were conducted to further characterize the oncogenic impact of Gas6/Axl autocrine signaling in J82 cells. Tumorigenesis was slow and tumor volumes varied widely relative to many well-studied tumor cell lines from other cancers, to an extent that drug efficacy studies were impractical. Hypothesizing that Gas6/Axl autocrine signaling in tumor xenografts might over time lead to a more aggressive tumor phenotype, 15 mice were implanted with $10^6$ cells each; when tumors reached 300 mm$^3$ they were excised, and tumor fragments were serially passaged in mice for 3 x 50-day tumor re-growth cycles. Although some later-cycle individual J82-derived tumors grew faster than others as measured by volume, these were frequently hemorrhagic and necrotic. Tumor tissue samples were taken at each cycle and 14 new cell lines were derived from tumor explants. None of these cell lines formed tumors faster than J82: the growth profiles of 10 J82 tumors (Fig 8A, left) and 10 tumors from a representative J82 tumor derived cell line (MDXC1, Fig 8A, center) showed wide variability yet similar mean growth rates (Fig 8A, right). Quantitative RT-PCR analysis of 8 tumor xenograft-derived cell lines showed that none had lost expression of either *GAS6* or *AXL*; in fact, 6 of 8 had acquired significantly increased expression of *MERTK* (also activated by Gas6) and *KDR* (Fig 8B). Consistent with high *MERTK* transcript levels, MERTK protein abundance in

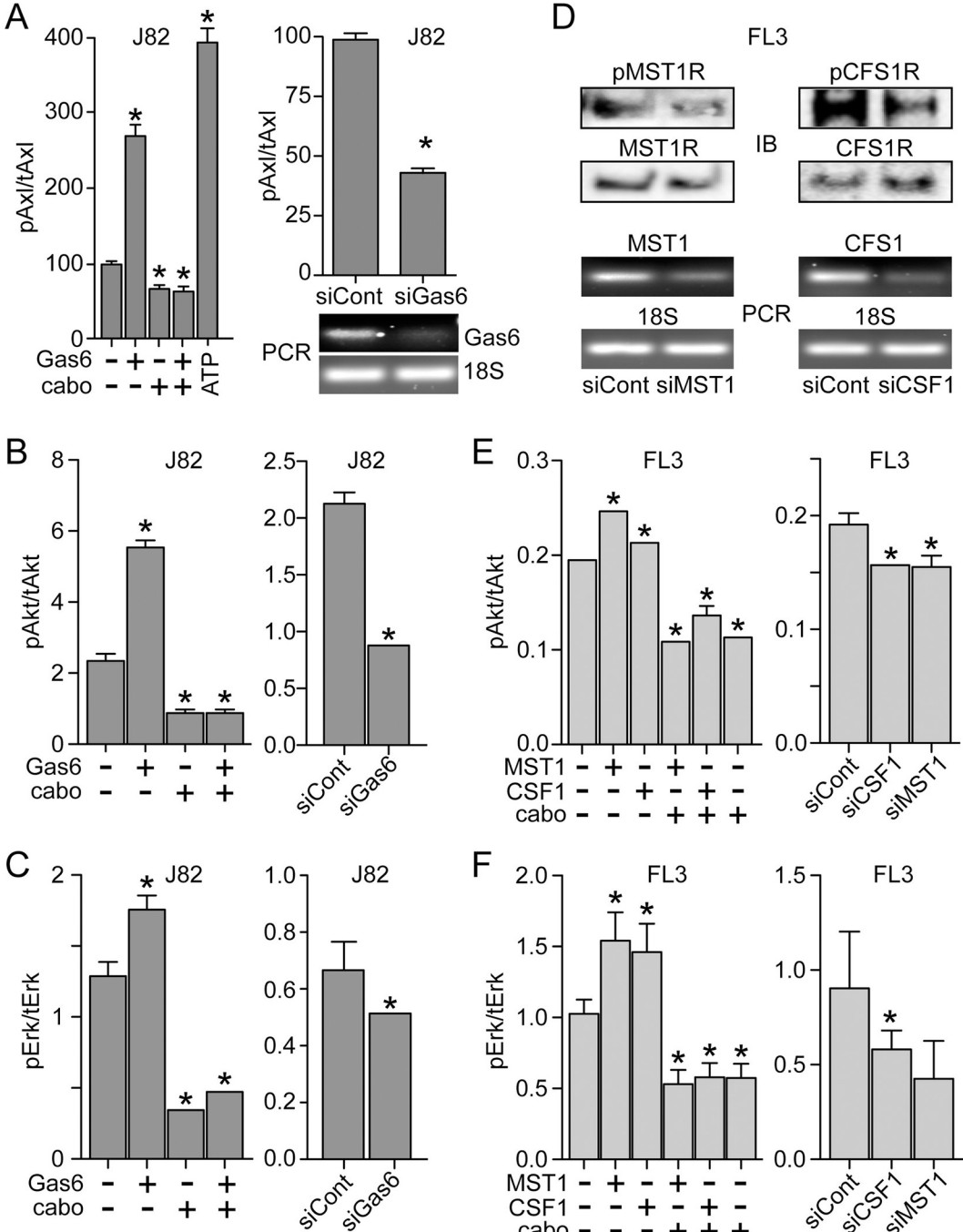

**Fig 5. Autocrine RTK signaling in BCa cell lines. (A)** Left: Axl kinase activation (phospho-Axl/total-Axl protein, pAxl/tAxl), as determined by 2-site immunoassay, in serum-deprived J82 cells treated with Gas6 (5.7 nM, 1 h at 37˚C) in the presence or absence of cabozantinib (cabo; 300 nM), relative to untreated cells. Maximum possible Axl kinase activation was measured by adding ATP during the immunoassay (20 micromolar, 20 min at 25˚C) of lysates prepared from untreated J82 cells. Right top: Axl kinase activation (pAxl/tAxl) in serum-deprived J82 cells that had been transfected with GAS6 siRNA or scrambled (control) siRNA. In both panels, values are the mean +/- SD from triplicate samples; asterisks indicate significant difference from untreated control (p < 0.05). Right bottom: ethidium bromide visualization of PCR products specific for GAS6 mRNA (upper panel) vs 18S rRNA (lower panel) resolved by agarose gel electrophoresis. **(B)** Left: Akt kinase activation (pAkt/tAkt), as determined by 2-site immunoassay, in serum-deprived J82 cells treated with Gas6 (5.7 nM, 1 h at 37˚C) in the presence or absence of cabozantinib (cabo; 300 nM), relative to untreated cells. Right: Akt kinase activation (pAkt/tAkt) in serum-deprived J82 cells that had been transfected with Gas6 siRNA or scrambled (control) siRNA. In both panels, values are the mean +/- SD from triplicate samples; asterisks indicate significant difference from untreated control

(p < 0.05). **(C)** Left: Erk kinase activation (pErk/tErk), as determined by 2-site immunoassay, in serum-deprived J82 cells treated with Gas6 (5.7 nM, 1 h at 37˚C) in the presence or absence of cabozantinib (cabo; 300 nM), relative to untreated cells. Right: Erk kinase activation (pErk/tErk) in serum-deprived J82 cells that had been transfected with GAS6 siRNA or scrambled (control) siRNA. In both panels, values are the mean +/- SD from triplicate samples; asterisks indicate significant difference from untreated control (p < 0.05). **(D)** Upper panels: Immunoblot (IB) of phospho-MST1R (pMST1R) and MST1R (left) or pCSF1R and CSF1R (right) in lysates from serum-deprived FL3 cells after transfection with siRNA directed against MST1 (siMST1, left) or CSF1 (siCSF1, right) or control siRNA (siCont). Lower panels: Ethidium bromide visualization of PCR products (PCR) specific for MST1 (left) or CSF1 (right) mRNA or 18S RNA resolved by agarose gel electrophoresis. **(E)** Left: Akt kinase activation (pAkt/tAkt) in serum-deprived FL3 cells treated with MST1 (5 nM) or CSF1 (10 nM) in the presence or absence of cabozantinib (cabo; 300 nM), relative to untreated cells. Right: Akt kinase activation (pAkt/tAkt) in FL3 cells that had been transfected with siRNA directed against CSF1 or MST1, or with control siRNA. In both panels, values are the mean +/- SD from triplicate samples; asterisks indicate significant difference from control (p < 0.05). **(F)** Left: Erk kinase activation (pErk/tErk) in serum-deprived J82 cells treated with MST1 or CSF1 in the presence or absence of cabozantinib (cabo), relative to untreated cells. Right: Erk kinase activation (pErk/tErk) in FL3 cells that had been transfected with siRNA directed against CSF1 or MST1, or with control siRNA. In both panels, values are the mean +/- SD from triplicate samples; asterisks indicate significant difference from untreated control (p < 0.05). All results are representative of at least 3 experiments.

MDXC1 cells was several-fold higher than that of J82 (Fig 8C). Exogenous Gas6 stimulated robust MERTK autophosphorylation, and the MERTK inhibitor TP0903 reduced phospho-MERTK content below that of untreated serum-deprived cells, suggestive of autocrine Gas6/MERTK activation (Fig 8C). J82 and MDXC1 cells grew at similar rates in 2D culture, but MDXC1 was notably less motile than J82, at rest or with Gas6 stimulation (Fig 8D).

Although persistent *AXL/GAS6* co-overexpression and acquired *MERTK* and *KDR* overexpression did not accelerate tumor xenograft growth rate, other concurrent phenotypic changes might nonetheless bear on disease progression. Seeking an overview of such changes, we used the Nanostring PanCancer Progression panel to analyze mRNA expression in J82 and MDXC1. Of the 770 genes in this panel (identified in S4A Table in S1 File), 321 were expressed at significantly different levels in MDXC1 relative to J82 (Qlucore 2-group comparison p = 0.021, q = 0.05; Fig 8E; S4B Table in S1 File). Ingenuity Pathway Analysis [40] of the 321 genes revealed that the top significantly matched *Diseases and Bio-functions* included several categories of cell migration, cell death and cell proliferation. Consistent with experimental observations, the z-scores indicated inactivated migration and proliferation in MDXC1 relative to J82 and activated cell death in MDXC1 relative to J82 (S5A Table in S1 File). Other notable findings in the IPA analysis were consistent with a transition from inflammatory AXL signaling to tolerogenic MERTK signaling, as described previously by Zagorska et al. [43]. MDXC1 expression of several proinflammatory mediators (e.g., *CXCL8*, *IL1A*, *IL1B*, *IL11*, *IL6*, *IL18*, and *PLAU*, S4B Table in S1 File) was significantly diminished while negative regulators of inflammation were significantly increased (e.g., *SPARC*, *TGF-β1*) relative to J82, and top IPA predicted *Upstream Regulators* that MDXC1 acquired were transforming growth factor beta-1, estradiol and dexamethasone (S5B Table in S1 File), the latter a known inducer of *MERTK* expression [43]. Top IPA predicted *Canonical Pathways* were also consistent with this phenotypic transition (S5C Table in S1 File).

## Discussion

Among several important discoveries, the comprehensive molecular analyses of BCa patient tumor samples published to date divided profound tumor heterogeneity into distinct phenotypes [14–22]. Although these broad classes display only a few dominant oncogenic pathways recognized in other cancers, they facilitate a process of systematic refinement of molecular pathogenesis that, for BCa in particular, is challenged with proving the criticality of less frequent, more complex and potentially transient processes driving disease progression.

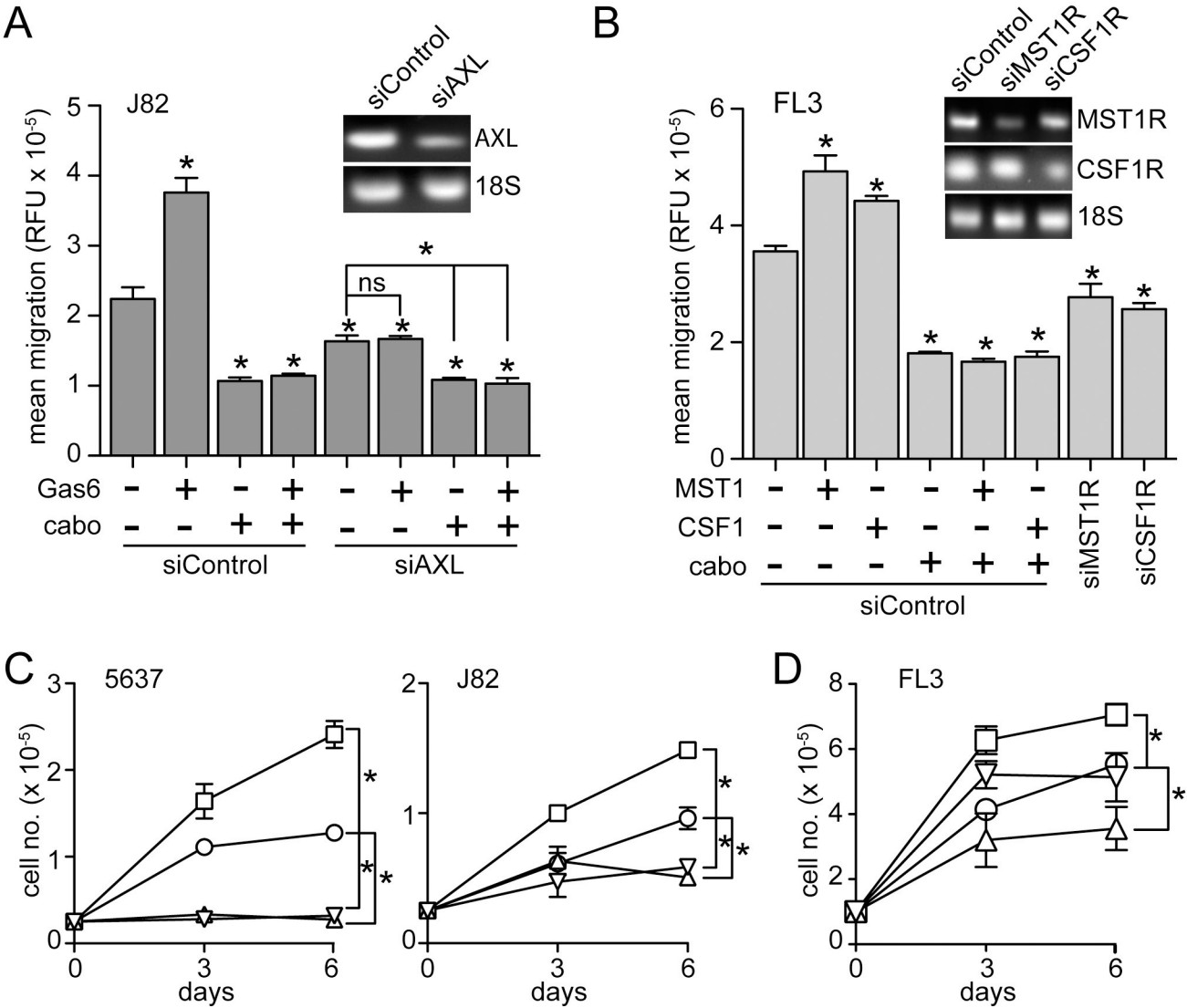

**Fig 6. Autocrine RTK driven migration and proliferation in BCa cell lines. (A)** Migration (16 h, 37˚C) of serum deprived J82 cells previously transfected with siRNA directed against AXL (siAXL) or control siRNA (siControl), treated with Gas6 (5.7 nM) in the presence or absence of cabozantinib (cabo; 300 nM), relative to untreated cells. Values are the mean +/- SD from triplicate samples; asterisks indicate significant difference from control ($p < 0.05$; ns, not significant). All results are representative of at least 3 experiments. Inset: ethidium bromide visualization of PCR products specific for AXL mRNA (upper panel) or 18S rRNA (lower panel) resolved by agarose gel electrophoresis. **(B)** Migration (16 h, 37˚C) of serum deprived FL3 cells previously transfected with siRNA directed against MST1R (siMST1R), CSF1R (siCSF1R) or control siRNA (siControl), treated with MST1 (5 nM) or CSF1 (10 nM) in the presence or absence of cabozantinib (cabo; 300 nM), relative to untreated cells. Values are the mean +/- SD from triplicate samples; asterisks indicate significant difference from untreated control ($p < 0.05$). All results are representative of at least 3 experiments. Inset: ethidium bromide visualization of PCR products specific for MST1R, (upper panel), CSF1R (middle panel), or 18S RNA (lower panel) resolved by agarose gel electrophoresis. **(C)** Proliferation (cell no. x $10^{-5}$, 6 d, 37˚C) of 5637 (left) or J82 (right) cells in 1% FBS left untreated (control; circles) or treated with Gas6 alone (5.7 nM; squares), cabozantinib alone (300 nM, triangles) or Gas6 and cabozantinib (inverted triangles). Media with additives was replaced on days 1, 2, and 4; cells were counted on days 0, 3 and 6. Values are the mean +/- SD from triplicate samples; asterisks indicate significant difference from control ($p < 0.05$). **(D)** Proliferation (cell no. x $10^{-5}$, 6 d, 37˚C) of FL3 cells in 1% FBS left untreated (control; circles) or treated with MST1 alone (5 nM; squares), cabozantinib alone (300 nM, triangles) or MST1 and cabozantinib (inverted triangles). Media replacement and cell counting as per panel C. Values are the mean +/- SD from triplicate samples; asterisks indicate significant difference from control ($p < 0.05$).

Accordingly, our focus was averted from well-studied oncogenic protein TK alterations, such as those affecting fibroblast growth factor receptors [11, 12, 14, 21, 44], and directed toward alterations that are as yet only circumstantially implicated in malignancy, with the goals of

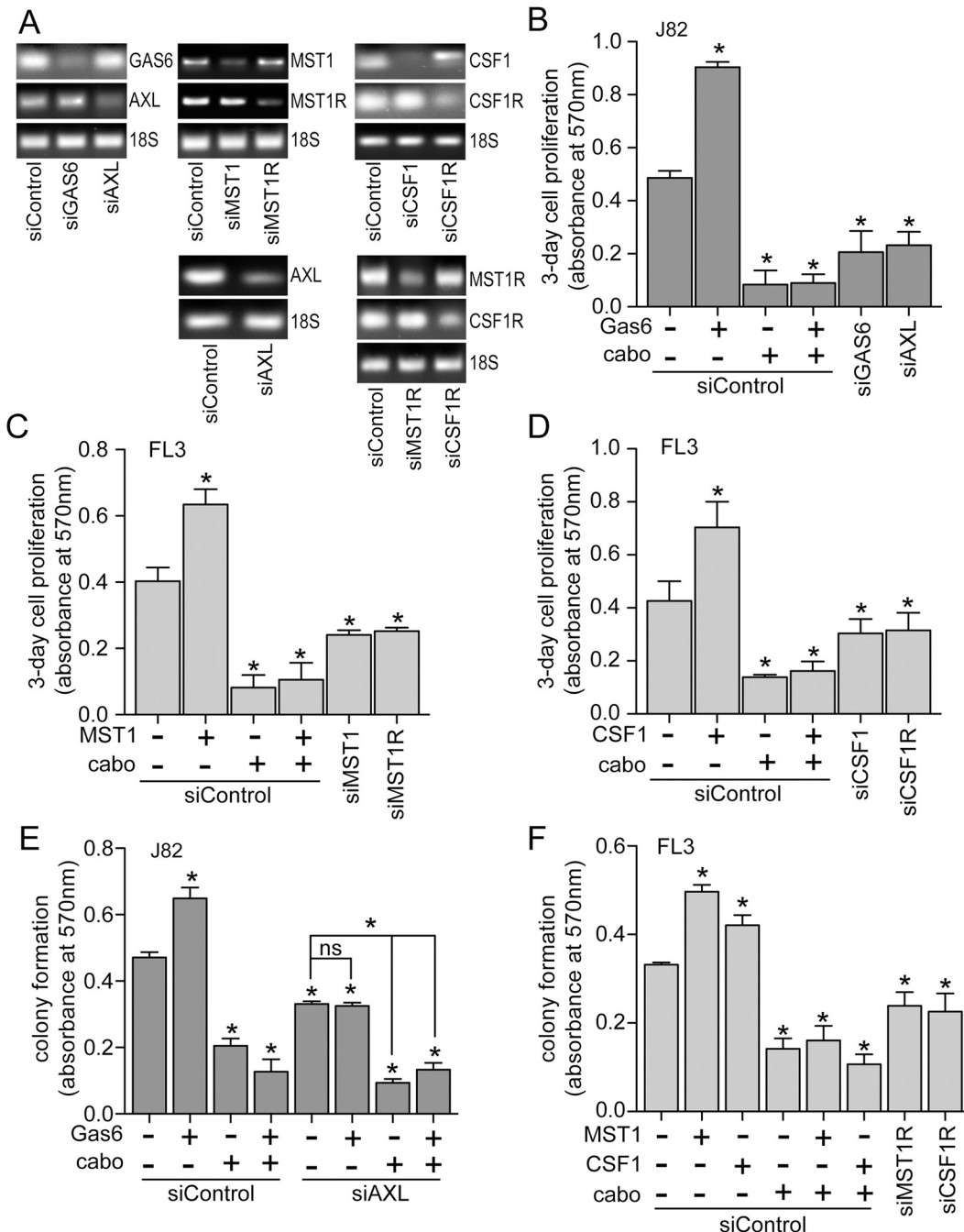

**Fig 7. Autocrine RTK driven proliferation and colony formation in BCa cell lines. (A)** PCR products showing siRNA suppression (upper and middle panels in each group) vs. 18S rRNA loading controls (lower panel in each group) resolved by agarose gel electrophoresis and visualized using ethidium bromide staining, from cell lines used for results shown in panels B–F. Upper left: J82 cells transfected with siRNA directed against GAS6 or AXL used for results shown in panel B. Upper middle: FL3 cells transfected with siRNA directed against MST1 or MST1R used for results shown in panel C. Upper right: FL3 cells transfected with siRNA directed against CSF1 or CSF1R used for results shown in panel D. Lower left: J82 cells transfected with siRNA directed against AXL used for results shown in panel E. Lower right: FL3 cells transfected with siRNA directed against MST1R or CSF1R used for results shown in panel F. **(B)** Proliferation (abs at 570 nm, 3 d, 37°C) of J82 cells in 1% FBS previously transfected with siRNA directed against GAS6 (siGAS6), AXL (siAXL) or control siRNA (siControl) that were left untreated or treated with Gas6 (5.7 nM) in the absence or presence of cabozantinib (cabo, 300 nM). **(C)** Proliferation of FL3 cells in 1% FBS previously transfected with siRNA directed against MST1 (siMST1), MST1R (siMST1R) or control siRNA (siControl) that were left untreated or treated with MST1 (5 nM) in the absence or presence of cabozantinib (cabo, 300 nM). **(D)** Proliferation of FL3 cells in 1% FBS previously transfected with siRNA directed against

CSF1 (siCSF1), CSF1R (siCSFR) or control siRNA (siControl) that were left untreated or treated with CSF1 (10 nM) in the absence or presence of cabozantinib (cabo, 300 nM). **(E)** Anchorage independent growth (abs at 570 nm) of J82 cells in 1% FBS previously transfected with siRNA directed against AXL (siAXL) or control siRNA (siControl) that were left untreated or treated with Gas6 (5.7 nM) in the absence or presence of cabozantinib (cabo, 300 nM). **(F)** Anchorage independent growth of FL3 cells in 1% FBS previously transfected with siRNA directed against MST1R (siMST1R), CSF1R (siCSFR) or control siRNA (siControl) that were left untreated or treated with MST1 (5 nM) or CSF1 (10 nM) in the absence or presence of cabozantinib (cabo, 300 nM). In panels C—F, values are the mean +/- SD from triplicate samples; asterisks indicate significant difference from untreated control ($p < 0.05$, ns, not significant). In all panels, data are representative of 3 or more independent experiments.

using TK inhibitors to their best therapeutic advantage and identifying new TK pathways for further study. We surveyed data obtained from 408 MIBC tumor samples previously classified into broad molecular phenotypes with distinct median overall survival periods (N, BS, L, LI and LP; [22]) for potentially oncogenic TK alterations, such as amplification and overexpression that might underlie catalytic overactivity, and for coincident overexpression of receptor TKs and their cognate ligands. Prompted by the positive results of a clinical trial of cabozantinib in patients with advanced or metastatic BCa [23], we further focused on RTKs targeted by this multikinase inhibitor for aberrant features and phenotypic distribution.

The expression profiles of 89 TKs analyzed across N, BS, L, LI and LP phenotypes fell into 3 broad groups: 2 with reciprocal patterns of overexpression among poorer surviving N and BS phenotypes and the best surviving LP phenotype (TK groups 1 and 2, respectively), and a group of 37 TKs with expression patterns that were independent of molecular phenotype (TK group 3). The significant differential distribution of group 1 and 2 TKs across phenotypes indicates that among the >3000 genes used to develop the phenotypes [22], these 52 TKs were defining elements, which in turn implies possible functional contributions to clinical distinctions, including survival. Although the combined incidence of potentially oncogenic mRNA overexpression and/or gene amplification events in all 3 groups were similar, significant co-occurrence of these alterations was disproportionately higher in group 1, in both number of samples affected and fraction of TK pairings. OS and PFS for patients harboring these alterations in group 1 TKs were significantly worse than for those without alterations, consistent with the reported relatively poor survival of N and BS phenotypes [22]. In contrast, patients with the same alterations in group 2 TKs had significantly better OS and DFS than the unaltered group, again consistent with more frequent occurrence in the better surviving L and LP phenotypes [22]. Patients harboring overexpression and/or amplification of group 3 TKs also showed significantly diminished DFS relative to those without alteration. These findings reinforce the likelihood that among all TKs, overexpression and/or amplification of groups 1 and 3 TKs contribute disproportionately to BCa progression.

The most recent Cabometyx$^{TM}$ (cabozantinib) prescribing information available from the US Food and Drug Administration [36] identifies AXL, FLT1, FLT3, FLT4, KDR, KIT, MET, MERTK, NTRK2, RET, ROS1, TYRO3 and TEK as its targets. We found that cabozantinib inhibited these RTKs *in vitro* with $IC_{50}$ values <50 nM, with the exception of KIT ($IC_{50}$ 143 nM), FLT1 ($IC_{50}$ 333 nM) and TEK (not included in our assays). Yakes et al. [31] reported potent TEK inhibition *in vitro* ($IC_{50}$ 14 nM), but less potent MST1R inhibition than we observed ($IC_{50}$ 124 nM *vs.* 42 nM). Niehus *et al.* [35] reported complete inhibition of CSF1R activation in intact cells, consistent with potent inhibition we observed *in vitro* ($IC_{50}$ 14 nM). In addition to these 16 targets, we observed potent inhibition of 3 other RTKs *in vitro*: DDR1, DDR2 and NTRK3, with $IC_{50}$ values of 11, 0.5, 20 and 3 nM, respectively.

Combined, these 19 ctRTKs were overexpressed and/or amplified in most (69%) MIBC cases in the TCGA dataset analyzed by Robertson et al. [22], with individual alteration

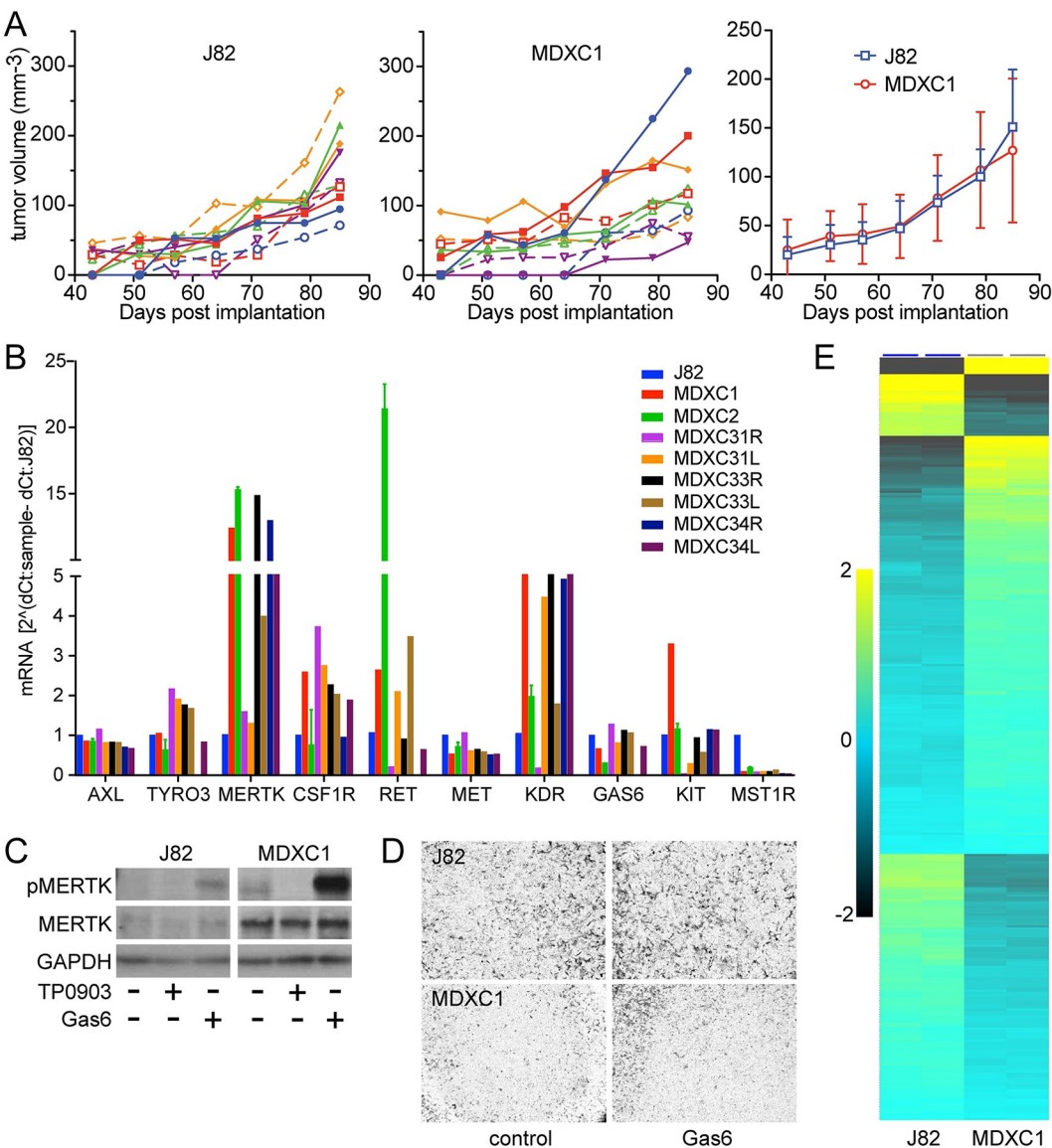

**Fig 8. Serial passage of J82 cell xenografts in mice. (A)** Tumor volume (mm³) of individual J82 (left) or MDXC1 (middle) cell xenografts in mice, or mean (+/- SD) tumor growth of both cell line xenografts (right), vs. days post implantation. (B) mRNA expression levels of *AXL, TYRO3, MERTK, CSF1R, RET, MET, KDR, GAS6, KIT* and *MST1R* in J82 cells and 8 J82 tumor-derived cell lines (MDXC1, MDXC2, MDXC31R, MDXC31L, MDXC33R, MDXC33L, MDXC34R and MDXC34L) as determined by qRT-PCR. **(C)** Immunoblot of lysates from J82 and MDXC1 lysates for phospho-MERTK (pMERTK, upper panels), MERTK (middle panels) or GAPDH (lower panels); intact cells had been treated prior to lysis with Gas6 (5.7 nM) or TP0903 (300 nM) as indicated. Results are representative of 3 or more independent experiments. **(D)** Light micrographs (10x objective magnification) of stained J82 (upper panels) of MDXC1 (lower panels) 24 h cell migration in the absence (control, left) or presence of added Gas6 (Gas6, 5.7 nM, right). **(E)** Heat map showing relative mRNA expression levels of 321 genes of the 770 gene Nanostring PanCancer Progression Panel (clustered hierarchically) that were significantly different among untreated J82 (left) and MDXC1 (right) cells (two-group comparison p = 0.021, q = 0.05; S5 Table in S1 File).

frequencies from 4% to 20%. Co-occurring alterations were also frequent (150 events). Sixteen ctRTKs were in TK groups 1 and 3, whose combined alterations occurred disproportionately in patients with diminished survival relative to those without alteration; indeed, the patient cohort harboring alterations in 14 ctRTKs (all except DDR1, FLT1/4, KIT and MST1R; 46% of cases) had significantly worse OS and DFS than those without alterations. The combination of

wide target spectrum and high frequency of target alteration implicated in worse outcome is consistent with the relatively high overall response rate (33%) observed in our clinical trial of cabozantinib in advanced BCa patents [23]. However, wide target spectrum also poses serious challenges to developing diagnostic biomarker panels and learning the molecular pathogenesis underlying individual patient responses. Anticipating that BCa-derived cell lines would be invaluable in that process, a preliminary characterization of how their TK expression profiles resembled those of TCGA samples was performed to help optimize their use in assessing the criticality of specific TK profiles in oncogenically relevant bioactivities.

We determined mRNA copy number for 31 TKs in 15 BCa-derived cell lines; this TK set represented groups 1 and 2 as defined using TCGA samples and retained the same significant differential distribution across LP vs. (LI + L) vs. (BS + N) phenotypes. A simple method was used to classify these cell lines among these molecular phenotypes on the basis of TK expression profile alone. We noted that several features discussed below, such as concurrent overexpression of RTK/cognate ligand pairs, and/or EMT transcriptional activators, occurred more frequently in cell lines classified as N or BS, corresponding to patient groups with worse survival relative those of other phenotypes. The classifications helped in selecting cell lines for studies of autocrine signaling, but may have more general utility, e.g., in determining the effects of multikinase inhibitors when specific combinations of active target pathways are present. Classification may also help to integrate kinomic with other -omic analyses to better inform cell line selection for studies aimed at specific BCa patient cohorts. We note that our classification is in reasonable agreement with those of Earl et al. [45], despite substantial differences between our methods.

Prevalent mechanisms of oncogenic TK activation (i.e., gene amplification, chromosomal rearrangements, gain of function mutations, and autocrine activation or aberrant ligand production in the tumor microenvironment) suppress immune surveillance and enhance tumor cell survival, proliferation, motility and invasion in a wide spectrum of cancers and thereby drive tumor progression, metastasis and drug resistance [46–48]. Among these mechanisms, autocrine activation and aberrant ligand production in the tumor microenvironment may appear as co-expression in RNA Seq analysis of tumor samples and occur with relatively low incidence. Both are, nonetheless, capable disease drivers and drug resistance pathways.

We found significant co-occurrence of mRNA overexpression (>2-fold RNASeq V2 z-scores) and/or gene amplification of *AXL*, *CSF1R*, *DDR2*, *KDR*, *MST1R*, *PDGFRA* and *TEK* and their cognate ligands among 65/408 cases (16%) in the BCa TCGA dataset [22]. These findings indicate that oncogenic autocrine RTK signaling could have occurred and provide impetus to fully determine its extent and criticality in BCa. RTK/ligand co-expression was confirmed by immunoblot analysis of CSF1R/CSF1, AXL/GAS6, and MST1R/MST1 in 9 BCa cell lines, reinforcing the hypothesis that autocrine signaling occurs in BCa patients. Six co-altered RTK/ligand pathways found are targeted by cabozantinib, and cabozantinib suppressed proliferation, motility and anchorage independent growth below basal levels in BCa cell lines harboring these alterations. All but 2 RTK/ligand pairs (*KDR/VEGFA* and *MST1R/MST1*) affected group 1 TK pathways that were frequently altered in the lower survival of N and BS phenotypes. Each co-alteration was manifested in <3% of cases, so although median OS and DFS values of patients harboring these events were below those of unaltered cases, they were not so extreme as to meet a 5% significance threshold. The fact that log-rank p values approached that threshold when group 1 cases were combined supports the possibility that these events were clinically relevant. Median OS and DFS values of patients harboring the group 2 MST1R/MST1 co-alteration, in contrast, segregated with the more favorable survival of the L and LP phenotypes. Whether this co-alteration has lower oncogenic potential than group 1 RTK/ligand pairs, or other features in the L and LP phenotypes offset its impact, remains to be determined.

Epithelial-to-mesenchymal transition (EMT) often precedes, and in model systems has been shown to promote, tumor invasiveness and metastasis [49]. Concurrent overexpression of TKs and a set of EMT TAs that they regulate was used as a proxy of oncogenic TK pathway activation in samples from the BCa TCGA dataset [22]. We also measured TA mRNA copy number in the 15 BCa-derived cell lines used for RTK analysis. Heat map analysis of TCGA and cell line data revealed a clear pattern of higher expression for *MYC*, *RUNX2*, *SNAI1*, *SNAI2*, *SOX2*, *SOX9*, *TWIST1*, *ZEB1* and *ZEB2* which strongly resembled the pattern of group 1 TKs, consistent with oncogenic impact. Indeed, co-occurring alterations among group 1 TKs and EMT TAs were 3-fold more frequent than for group 2 TKs, and 24% of those were associated with significantly lower OS and/or PFS relative to unaltered cases. mRNA overexpression and/or gene amplification among these EMT TAs occurred in a combined 174 (43%) TCGA samples, 59 (34%) of which had significantly co-occurring EMT TA alterations. OS for patients with co-occurring EMT TA alterations was 27.43 mos. *vs*. 44.91 mos. for those without alterations (p = 0.0175). The possibility that co-occurring alterations of RTK/ligand pairs or TKs and EMT TAs detected in tumor/liquid biopsies, plasma DNA or exosomes from BCa patients might predict clinical outcome warrants further study.

Direct evidence of autocrine signaling by AXL, CSF1R and MST1R pathways was found in BCa-derived cell lines. Selective siRNAs directed against ligand or RTK suppressed receptor and canonical downstream mediator activation, as well as cell proliferation, migration and anchorage-independent growth, below basal levels. The results obtained using selective siRNAs support the concept that significantly reduced basal levels of the same biochemical proxies and biological activities by cabozantinib in these cell lines also occurred by blocking autocrine RTK signaling, and suggest that this mechanism of action underlies a portion of the efficacy we observed in BCa patients on a phase II cabozantinib clinical trial [23], and may be an important disease driver in BCa. While this manuscript was in preparation, an independent report [50] found evidence of GAS6 expression in BCa tissue samples associated with lower survival, and functional impact of GAS6 signaling in 5 BCa-derived cell lines; those findings reinforce the likelihood that autocrine AXL signaling contributes to BCa disease progression in a subset of patients.

Serial passage of the human BCa-derived cell line J82 as tumor xenografts in mice generated several J82-derived cell lines. One such line, MDXC1 grew comparably to J82 in 2D culture, but was less motile and displayed significant expression changes in cancer related genes. Notably, none of the J82 xenograft-derived cell lines analyzed had lost expression of either *GAS6* or *AXL*, suggesting that retention of this activated pathway contributed to tumorigenicity. Six of 8 derived cell lines acquired significantly higher expression of *MERTK* and *KDR*, and MDXC1 cells showed evidence of robust MERTK activation. Ingenuity Pathway Analysis [40] of 321 differentially expressed genes in the Nanostring PanCancer Progression 770 gene panel revealed that in MDXC1, genes comprising cell migration and cell proliferation bio-functions were predominantly inactivated, while those comprising cell death were activated. Accompanying this were significant changes in regulators of inflammation that were consistent with a shift from AXL to MERTK signaling, which have been distinguished in carefully controlled studies [43] that refine our understanding of TAM receptor signaling in tumor immune surveillance [51, 52]. Indeed, recent studies show that MERTK signaling contributes to an immunosuppressive environment by inducing an anti-inflammatory cytokine profile and regulating checkpoint inhibitor signaling in hematopoietic and solid tumors [53–55]. Our xenograft studies in immunocompromised SCID/Beige mice would be unlikely to display these changes as a pro-tumorigenic transition from J82 to MDXC1. In humans, however, such a transition might lead to more effective tumor evasion of immune recognition and disease progression, despite increased tumor cell death and neo-antigen abundance. We recently reported that in patients

with advanced BCa, cabozantinib treatment modulated immune checkpoint regulators PD-1 and TIM-3, increased anti-tumor monocytes, and decreased pro-tumorigenic monocytes and myeloid-derived suppressor cell (MDSC) populations; a greater-than-median decrease in the granulocytic MDSC population was significantly associated with a favorable outcome [23]. While our focus here was on coincident overexpression or amplification of GAS6, AXL and MERTK as proxies of putative driver roles and relevant cabozantinib targets, our findings also suggest that a more comprehensive characterization of these pathways as drivers in tumor and immune compartments should inform efforts to optimally combine TKIs and checkpoint inhibitors for BCa treatment.

## Supporting information

**S1 File.**
(XLSX)

## Author Contributions

**Conceptualization:** Young H. Lee, Robert L. Grubb, III, Piyush K. Agarwal, Andrea B. Apolo, Donald P. Bottaro.

**Data curation:** Young H. Lee, Molly M. Lee, Dinuka M. De Silva, Arpita Roy, Cara E. Wright, Tiffany K. Wong, Rene Costello, Oluwole Olaku, Andrea B. Apolo, Donald P. Bottaro.

**Formal analysis:** Young H. Lee, Molly M. Lee, Dinuka M. De Silva, Arpita Roy, Cara E. Wright, Tiffany K. Wong, Rene Costello, Oluwole Olaku, Andrea B. Apolo, Donald P. Bottaro.

**Funding acquisition:** Donald P. Bottaro.

**Investigation:** Young H. Lee, Molly M. Lee, Dinuka M. De Silva, Arpita Roy, Cara E. Wright, Tiffany K. Wong, Rene Costello, Oluwole Olaku, Piyush K. Agarwal, Andrea B. Apolo, Donald P. Bottaro.

**Methodology:** Young H. Lee, Molly M. Lee, Dinuka M. De Silva, Arpita Roy, Cara E. Wright, Tiffany K. Wong, Rene Costello, Oluwole Olaku, Andrea B. Apolo, Donald P. Bottaro.

**Project administration:** Andrea B. Apolo, Donald P. Bottaro.

**Resources:** Piyush K. Agarwal, Andrea B. Apolo, Donald P. Bottaro.

**Software:** Donald P. Bottaro.

**Supervision:** Andrea B. Apolo, Donald P. Bottaro.

**Validation:** Young H. Lee, Molly M. Lee, Dinuka M. De Silva, Arpita Roy, Cara E. Wright, Oluwole Olaku, Andrea B. Apolo, Donald P. Bottaro.

**Visualization:** Young H. Lee, Molly M. Lee, Dinuka M. De Silva, Arpita Roy, Cara E. Wright, Andrea B. Apolo, Donald P. Bottaro.

**Writing – original draft:** Young H. Lee, Molly M. Lee, Dinuka M. De Silva, Arpita Roy, Andrea B. Apolo, Donald P. Bottaro.

**Writing – review & editing:** Young H. Lee, Molly M. Lee, Dinuka M. De Silva, Arpita Roy, Cara E. Wright, Rene Costello, Robert L. Grubb, III, Piyush K. Agarwal, Andrea B. Apolo, Donald P. Bottaro.

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
