## [Decision Letter · Decision Letter 0]

1 Mar 2021

PONE-D-20-33596

Autocrine Signaling by Receptor Tyrosine Kinases in Urothelial Carcinoma of the Bladder

PLOS ONE

Dear Dr. Bottaro,

Thank you for submitting your manuscript to PLOS ONE. After careful consideration, we feel that it has merit but does not fully meet PLOS ONE’s publication criteria as it currently stands. Therefore, we invite you to submit a revised version of the manuscript that addresses the points raised during the review process.

Three experts in the field reviewed the present study. They consider positively its novelty and scientific interest. However, the authors should answer several criticisms pointed by the reviewers. Especially, authors should respond carefully about the technical aspects of the proposal. These include the lack of phosphorylation measurements and the possibility of gene amplifications as a causative mechanism. Other points refer to the clarity of the presentation, in particular regarding the way the figures are presented.

We look forward to receiving your revised manuscript.

Kind regards,

Pablo Garcia de Frutos

Academic Editor

PLOS ONE

Journal Requirements:

2. Thank you for agreeing previously to compile the sequence table of primers and siRNAs during the period of finding reviewers and peer review. As your manuscript has now been reviewed, could you please provide this table as a supplementary file?

3.PLOS ONE now requires that authors provide the original uncropped and unadjusted images underlying all blot or gel results reported in a submission’s figures or Supporting Information files. This policy and the journal’s other requirements for blot/gel reporting and figure preparation are described in detail at https://journals.plos.org/plosone/s/figures#loc-blot-and-gel-reporting-requirements and https://journals.plos.org/plosone/s/figures#loc-preparing-figures-from-image-files. When you submit your revised manuscript, please ensure that your figures adhere fully to these guidelines and provide the original underlying images for all blot or gel data reported in your submission. See the following link for instructions on providing the original image data: https://journals.plos.org/plosone/s/figures#loc-original-images-for-blots-and-gels.

5.We note that you have a patent relating to material pertinent to this article. Please provide an amended statement of Competing Interests to declare this patent (with details including name and number), along with any other relevant declarations relating to employment, consultancy, patents, products in development or modified products etc. Please confirm that this does not alter your adherence to all PLOS ONE policies on sharing data and materials, as detailed online in our guide for authors http://journals.plos.org/plosone/s/competing-interests by including the following statement: "This does not alter our adherence to  PLOS ONE policies on sharing data and materials.” If there are restrictions on sharing of data and/or materials, please state these. Please note that we cannot proceed with consideration of your article until this information has been declared.

Reviewers' comments:

Reviewer's Responses to Questions

**Comments to the Author**

1. Is the manuscript technically sound, and do the data support the conclusions?

Reviewer #1: Partly

Reviewer #2: Yes

Reviewer #3: Yes

2. Has the statistical analysis been performed appropriately and rigorously? 

Reviewer #1: Yes

Reviewer #2: I Don't Know

Reviewer #3: Yes

3. Have the authors made all data underlying the findings in their manuscript fully available?

Reviewer #1: Yes

Reviewer #2: No

Reviewer #3: Yes

4. Is the manuscript presented in an intelligible fashion and written in standard English?

Reviewer #1: Yes

Reviewer #2: Yes

Reviewer #3: Yes

5. Review Comments to the Author

Reviewer #1: In this work, Lee et al., explored the potential use of autocrine TK signaling as a suitable tool for managing advanced muscle invasive Bladder cancer. In particular, based on a previous trial using cabozantinib, they focused in potential Cabozantinib targets. Their analyses of TCGA database allow them to identify two groups of TK overexpressed (n= 31 and n=21) in opposite manner among the Basal/neuronal and luminal papillary tumors, whereas the remaining 37 TKs studied varied independently of molecular phenotype. They next explored mRNA expression of 31 TK gene in BCa cell lines and found that they can subclassify these cells into different molecular subtypes.All potential Cabozantinib targets displayed potentially oncogenic alterations in a large proportion of TCGA cases, and they also co-express high levels of their cognate ligands. These findings were corroborated in cell lines, in some cases also by immunoblot, suggesting possible autocrine signaling. Autocrine and ligand-dependent signaling, as well as its inhibition by cabozantinib, were confirmed in J82 and FL3 cells. Since they observed coincident overexpression of TKs and EMT transcriptional activators in TCGA, they studied whether autocrine signaling may mediate cell migration and this could be blocked by cabozantinib. In line with this autocrine signaling, xenografts of J82 cells serially passaged in mice showed persistent AXL/GAS6 co-overexpression and acquired MERTK and KDR overexpression.

The study is interesting as it may open new possible ways of treating advanced BC. However, a number of potential clarifications should be added prior acceptance.

1) The data regarding the phase II of cabozantinib in mUC should be discussed in terms of effectivity (also the authors should correct the reference of this study).

2) Why the authors excluded mutations in their first analysis of the TK genes?

3) How the TK genes to be analyzed were selected in cell lines?

4) How their assignment of cell lines to different BCa subtypes and their RNA data are correlated with those previously reported by Earl et al., 2015 (PMID: 25997541), including gene amplifications?

5) It is not clear whether the differences in clinical outcome between altered and non-altered groups are obtained for each molecular subtype or globally. They should clarify this point, given the intrinsic differences in clinical outcome among groups

6) In addition to EMT drivers, they should analyze whether the altered groups also showed enrichment of EMT signatures using GSEA. In the same line, patients harboring overexpression also displayed increased tendency to develop distant metastasis, either at diagnose or in follow up?

7) Number of cell lines were they confirm autocrine signaling is too short to generalize autocrine signaling in BC cell lines.

8) The metastatic capacity of J82 cells is increased upon serially passages in mice? Can it be blocked by cabozantinib?

Reviewer #2: The work is well done but the data is not presented in a manner by which the reader can follow. For example, in Figure 2, authors need to mark on the figure what is the difference between A, B, C, D. Western blots need to be redone - for example, that in Figure 4B upper three panels. In addition, the figure needs a loading control that is the same throughout the blot. Immunoblots in Figure 5D are cut too close to the band, and need to show empty space both before and after. Figure 5 - effect of adding ligands to the receptor or receptor phosphorylation not shown. Effect of genes overexpressing the receptor not shown.

Reviewer #3: Summary of submission: This work utilizes bladder cancer (BCa) data from The Cancer Genome Atlas (TCGA) database to postulate the role of receptor tyrosine kinases (RTKs) in BCa. Hypotheses are then tested using multiple human BCa cell lines. The results suggest that autocrine signaling by RTKs contributes to the malignant phenotypes of some of the BCa cell lines and that autocrine RTK signaling may be a good target for therapeutic intervention in BCa.

The scope of this work is wide ranging and includes both in silico and wet lab approaches. Overall, the work is well-designed and -performed. Two major and a few minor issues are noted that do not markedly reduce the enthusiasm for this work but that should be addressed prior to publication.

Major issues

1. The failure to directly measure RTK phosphorylation is noteworthy, particularly in those cases in which autocrine RTK signaling is inferred.

2. Similarly, the experiments do not appear to explicitly address the possibility that elevated RTK signaling could in some cases arise through RTK overexpression. This is of particular note given the attempt to identify the target(s) for carbozantinib.

Minor issues

1. It is unclear how the cell lines, RTKs, and ligands analyzed in Figure 4b were chosen, particularly given the apparent occasional incongruence between transcription data shown in Figure 4a and the protein expression data shown in Figure 4b.

2. Is there strong congruence between the cell line gene expression data reported in this work (Fig 3A and Fig 4A) and data available from the Broad CCLE? Similarly, if other biomarkers (from ref 21 cited in this work) characteristic of the N, BS, L, LI, and LP subtypes are applied to Broad CCLE data for the 15 BCa cell lines, are the subtype assignments the same as those reported in Table 1?

3. The authors’ attempts at brevity are greatly appreciated, particularly given the scope of this work. However, the paper appears to be a bit underwritten. Additional subheadings in the results section would improve readability. Likewise, additional transitioning between subsections and between and within paragraphs would improve clarity. For example, on page 14 (paragraph 1), the paper reads “cabozantinib inhibited 17 RTKs”. Yet, on page 14 (paragraph 2), the paper reads “All 19 ctRTKs displayed potentially oncogenic alterations”. It is not clear why 19 RTKs are being studied given that only 17 RTKs are inhibited by cabozantinib. Later, in paragraph 2 of page 14, the manuscript indicates that “patients harboring overexpression and/or amplification among 14 ctRTKs had significantly worse OS and DFS”. It is not apparent how these 14 RTKs were chosen.

4. Finally, the manuscript appears to occasionally use TK instead of RTK.

6. PLOS authors have the option to publish the peer review history of their article (what does this mean?). If published, this will include your full peer review and any attached files.

Reviewer #1: **Yes: **Jesus M Paramio

Reviewer #2: No

Reviewer #3: No

---

## [Author Response · Author response to Decision Letter 0]

20 May 2021

PONE-D-20-33596: "Autocrine Signaling by Receptor Tyrosine Kinases in Urothelial Carcinoma of the Bladder" by Young H. Lee et al. 

Response to Reviewers

Review Comments to the Author

Please note that all Reviewer comments are as received verbatim.

Reviewer #1:

In this work, Lee et al., explored the potential use of autocrine TK signaling as a suitable tool for managing advanced muscle invasive Bladder cancer. In particular, based on a previous trial using cabozantinib, they focused in potential Cabozantinib targets. Their analyses of TCGA database allow them to identify two groups of TK overexpressed (n= 31 and n=21) in opposite manner among the Basal/neuronal and luminal papillary tumors, whereas the remaining 37 TKs studied varied independently of molecular phenotype. They next explored mRNA expression of 31 TK gene in BCa cell lines and found that they can subclassify these cells into different molecular subtypes.All potential Cabozantinib targets displayed potentially oncogenic alterations in a large proportion of TCGA cases, and they also co-express high levels of their cognate ligands. These findings were corroborated in cell lines, in some cases also by immunoblot, suggesting possible autocrine signaling. Autocrine and ligand-dependent signaling, as well as its inhibition by cabozantinib, were confirmed in J82 and FL3 cells. Since they observed coincident overexpression of TKs and EMT transcriptional activators in TCGA, they studied whether autocrine signaling may mediate cell migration and this could be blocked by cabozantinib. In line with this autocrine signaling, xenografts of J82 cells serially passaged in mice showed persistent AXL/GAS6 co-overexpression and acquired MERTK and KDR overexpression.

The study is interesting as it may open new possible ways of treating advanced BC. However, a number of potential clarifications should be added prior acceptance.

Author Reply:

We thank Professor Paramio for his positive comments and careful review of our work.

1) The data regarding the phase II of cabozantinib in mUC should be discussed in terms of effectivity (also the authors should correct the reference of this study).

Author Reply:

We have revised the Introduction (pp. 4-5, lines 82-94) to more fully summarize the clinical trial results published in Lancet Oncology (our cited reference 23), their relationship to our prior study of BCa cell lines (our cited reference 24), and the rationale that these and other findings provided to identify the most likely oncogenic drivers among cabozantinib's TK targets. Our cited reference 23 also has been updated.

2) Why the authors excluded mutations in their first analysis of the TK genes?

Author Reply:

Although we are very interested in the impact of TK mutations in bladder cancer, several prior studies have described the most prevalent mutations and their predicted and/or observed oncogenic impact (our cited references 14-22). A comprehensive assessment of mutations across the entire TK family is also a big task for an individual lab (as opposed to a TCGA or similar large team collaboration), and because a substantial fraction of mutations would have unknown pathogenic impact and thus not bear comment, that portion of the work (often by a trainee) would be unpublishable. On the other hand, TK overexpression as a proxy for oncogenic impact in urothelial carcinoma of the bladder has not yet garnered as much attention from the research community. Because evidence of TK overexpression alone is not as strong as activating genomic alterations in guiding therapeutic strategy, we included additional proxies of pathway activation: coincident ligand overexpression (which might represent autocrine or local paracrine production, in view of how the TCGA samples were processed) and overexpression of EMT transcriptional activators that are known TK pathway effectors.

3) How the TK genes to be analyzed were selected in cell lines?

Author Reply:

We considered determining absolute mRNA copy number for all 52 group 1 and 2 TKs in the 15 BCa cell lines. There were challenges to this task, since we started with reliable primers for much smaller number of TKs. While expanding this primer set, we encountered TKs for which there were no published primers and/or readily available cell lines to serve as positive controls for primer testing. So, we assembled primer sets for group 1 and group 2 TKs in a 3:2 ratio (like the group of 52) to reach a subset that achieved the same correlation with TCGA patient molecular phenotype as the parent 52 TK set (p = 1.00 x10-4, q = 8.97 x 10-5, F2,405 > 9.42, R2 > 0.044 for the 3-group comparison F test: LP vs. (LI + L) vs. (BS +N)). This goal was met with 19 group 1 TKs and 12 group 2 TKs, or ~60% of the parent set. We have added text to the Results for clarification (p. 14, lines 369-374). While this strategy limited the cell line data modestly, it was a reasonable and practical way to provide relevant TK coverage and enable TK-based molecular phenotyping of most of the cell lines into the classes described by Robertson et al. (cited reference 22).

4) How their assignment of cell lines to different BCa subtypes and their RNA data are correlated with those previously reported by Earl et al., 2015 (PMID: 25997541), including gene amplifications?

Author Reply:

This is an interesting question and we thank the reviewer for calling the work of Earl et al. to our attention. The short answer is that there is reasonable agreement between our findings, despite the very different methods that were used.

To recap our method: each cell line TK PCR expression profile (n = 15, 31 TKs) was correlated with each TCGA sample TK RNASeq profile (n = 408, 31 TKs), the correlation coefficients were determined and averaged within each molecular phenotype defined by Robertson et al. (22), and cell lines were assigned to a phenotype by best average coefficient that was distinguished from coefficients for other phenotypes by t-test.

Earl et al. (2015) used two classification schemes, although both schemes matched cell line profiles to a molecular classifier signature set. From Earl et al. (verbatim): 

(1) “We applied the UBC molecular classifier based on gene expression defined by Sjodahl et al. [12] to identify lines most representative of the taxonomical groups proposed. Figure 6A shows that cell lines could be adscribed to the “Urobasal A”, “Urobasal B”, and “SCC-like” classifiers (Additional file 1: Table S12). The “Genomically Unstable” group was most commonly represented among the lines.”

(2) “Rebouissou et al. have recently reported on a 40-gene basal-like signature [25]. We have applied their 40-gene classifier to the cell line dataset and identify 4 major groups: lines with a predominant enrichment in the “Basal-like” signature; lines with a predominant enrichment in the “Non basal-like” signature; lines with enrichment of both signatures; and lines in which none of the signatures is enriched (Figure 6B and Additional file 1: Table S13).” 

The most important difference between our respective methods was that we used only TK expression data for developing correlation coefficients, whereas the signature sets used by Earl et al. were developed without restricting genes to any particular functional class.

Eleven of the 15 cell lines that we used were also studied by Earl et al. We adapt below Table S13 from Earl et al. (left side) to compare the classifications of these 11 cell lines (right side):

[PLEASE SEE THE ATTACHED FILE "Response to Reviewers.pdf" to view the table below in proper format]

Table S13. Pearson's correlations according to the 40-gene signature from Rebouissou et al. (Earl et al.) Comparison with Lee et al. (submitted)

Cell line Non basal-like MIBC Basal-like MIBC Lee et al. classification Passed t-test? Agreement w/ Earl?

5637 0.245 0.639 Basal squamous Yes Yes

HT1197 0.436 0.530 Basal squamous Yes Yes

HT1376 0.319 0.458 Luminal papillary Yes* No

J82 0.288 0.224 Basal squamous Yes No

RT112 0.502 0.614 Neuronal No Maybe*

RT4 0.695 0.373 Luminal papillary Yes Yes

SW-780 0.627 0.491 Luminal papillary Yes* Yes

T24 0.105 0.241 Basal squamous Yes Yes

TCCSUP 0.163 0.098 Basal squamous No No

UM-UC-3 0.173 0.152 Basal squamous Yes No

UM-UC-5 0.561 0.627 Luminal papillary Yes* No

Yes* = passed all t-tests except Luminal, the most closely related class.

Maybe* = Earl et al. had no neuronal classification, but RT112 had mixed but more basal phenotype by the Rebouissou et al. criteria.

Comparisons like this are interesting, but they have many caveats. For this reason we have chosen to only briefly mention the partial agreement of our findings and those of Earl et al. (Discussion, p. 30, lines 808-809) and we have added the Earl et al. citation to our cited references (#45, p. 38, lines 1028-1031).

5) It is not clear whether the differences in clinical outcome between altered and non-altered groups are obtained for each molecular subtype or globally. They should clarify this point, given the intrinsic differences in clinical outcome among groups

Author Reply:

Thank you for raising this point. We did not limit the Kaplan-Meier analyses to specific molecular phenotypes; in each case a global comparison was made among altered and unaltered gene groups. We have revised the Material and Methods text for clarification (p. 7, lines 157-160).

6) In addition to EMT drivers, they should analyze whether the altered groups also showed enrichment of EMT signatures using GSEA. In the same line, patients harboring overexpression also displayed increased tendency to develop distant metastasis, either at diagnose or in follow up?

Author Reply:

While we agree it would be interesting to determine whether any of the groups with lower survival also display enrichment of EMT signatures by GSEA compared to the unaltered groups, we feel that the amount of work involved is beyond the scope of this manuscript. We hope that an inspired reader of our work might consider the question and use the PLOS ONE online forum to post any findings of interest. 

We believe that correlating overexpression with metastasis will be of limited value, since the greatest single factor determining overall survival is metastatic disease. This implies that any cohort that shows reduced overall survival, regardless of the feature(s) used to define that group, is very likely to also show a shorter average time to metastasis.

7) Number of cell lines were they confirm autocrine signaling is too short to generalize autocrine signaling in BC cell lines.

Author Reply:

We did not intend to imply that autocrine signaling is a dominant feature among BCa cell lines in general. We show immunoblot evidence of co-expression of cognate ligand and receptor protein pairs in 9 BCa cell lines (some of which show ligand/receptor pairs for more than one RTK pathway) and functional evidence of autocrine signaling in J82 and FL3. In our summary of these findings at the end of the Introduction (p. 5, lines 107-110) we originally stated: “These findings reveal the prevalence and patterns of autocrine RTK signaling in BCa…” where we used the word “prevalence” to mean “frequency”; to avoid confusion we have substituted the word “frequency” in the revised text. In the Discussion (p. 32, lines 853-854) we originally stated: “Direct evidence of autocrine signaling by AXL, CSF1R and MST1R pathways was found in several BCa-derived cell lines.”; we have deleted the word “several” from this sentence in the revised manuscript to avoid any overstatement.

8) The metastatic capacity of J82 cells is increased upon serially passages in mice? Can it be blocked by cabozantinib?

Author Reply:

We did not observe that the metastatic capacity of J82 increased upon serial passages in mice and we do not know whether cabozantinib can block J82 metastasis in mice.

Reviewer #2:

The work is well done but the data is not presented in a manner by which the reader can follow. For example, in Figure 2, authors need to mark on the figure what is the difference between A, B, C, D. Western blots need to be redone - for example, that in Figure 4B upper three panels. In addition, the figure needs a loading control that is the same throughout the blot. Immunoblots in Figure 5D are cut too close to the band, and need to show empty space both before and after. Figure 5 - effect of adding ligands to the receptor or receptor phosphorylation not shown. Effect of genes overexpressing the receptor not shown.

Author Reply:

We thank the reviewer for their positive comments and careful review of our work.

We have revised Figure 2 to identify the groups being compared in each panel's symbol key. The comparisons in each panel are also defined in the figure's legend. 

We have replaced the CSF1R immunoblot image for RT4, RT112, J82, UMUC5 and UMUC3 cell lines Figure 4B. We have also added to this panel immunoblot detection results for DDR1, DDR2 and PTK7 proteins across the 9 BCa cell lines, as these are understudied and potentially oncogenic RTKs in urothelial carcinoma of the bladder. The Results text (p. 17, lines 465-473) and Figure 4 legend (p. 18, lines 487-489) have been revised accordingly. The loading control for Figure 4B was omitted in error and this has been corrected in the revised figure (panel B, bottom) and legend.

We wish to emphasize that we did not use immunoblotting here to quantitate protein content, but to simply detect the presence or absence of a protein product under the conditions specified. As shown in Figure 5, we use 2-site electrochemiluminescent immunoassays to quantitatively compare protein-to-phosphoprotein ratios under varying conditions. For example, we show that adding Gas6 to serum-deprived J82 cells significantly increased AXL phosphotyrosyl content (Fig 5A, compare bars for Gas6 "-" and "+" at left). This increase in receptor phosphorylation is reversed by cabozantinib treatment (Fig 5A, 3rd and 4th bars from the left). The remaining panels in Figure 5 show that addition of Gas6, MST1 or CSF1 stimulated significant increases in the phosphorylation of critical downstream effectors Akt and Erk.

Regarding the cropping of bands in Figure 5D, we believe that no information has been obscured by our figure production, and we provide public access to all full-size, uncropped images that are used for figures as required per PLOS ONE policy.

Reviewer #3:

Summary of submission: This work utilizes bladder cancer (BCa) data from The Cancer Genome Atlas (TCGA) database to postulate the role of receptor tyrosine kinases (RTKs) in BCa. Hypotheses are then tested using multiple human BCa cell lines. The results suggest that autocrine signaling by RTKs contributes to the malignant phenotypes of some of the BCa cell lines and that autocrine RTK signaling may be a good target for therapeutic intervention in BCa.

The scope of this work is wide ranging and includes both in silico and wet lab approaches. Overall, the work is well-designed and -performed. Two major and a few minor issues are noted that do not markedly reduce the enthusiasm for this work but that should be addressed prior to publication.

Author Reply:

We thank the reviewer for their positive comments and thorough review of our work.

Major issues

1. The failure to directly measure RTK phosphorylation is noteworthy, particularly in those cases in which autocrine RTK signaling is inferred.

Author Reply:

Please note that direct quantitative measurement of Axl tyrosyl phosphorylation is shown in Fig. 5A and MERTK tyrosyl phosphorylation is shown more qualitatively by immunoblot in Fig. 8C. Any perceived lack of evidence of RTK activation is not due to lack of expertise or diligence (the Bottaro lab has published extensively on TK phosphorylation), but rather by choice. We believe that today, RTK activation through tyrosyl phosphorylation is well-enough established to be assumed from evidence of ligand-stimulated biological effects (by ligand addition or specific pathway inhibition as shown in Figs. 5 – 8). Had we failed to find ligand-driven effects in experimental settings where they were expected, we certainly would have measured RTK activation as part of normal troubleshooting. Otherwise, we believe our effort is better used measuring pathway effects that are not yet established. 

2. Similarly, the experiments do not appear to explicitly address the possibility that elevated RTK signaling could in some cases arise through RTK overexpression. This is of particular note given the attempt to identify the target(s) for carbozantinib.

Author Reply:

We do appreciate the importance of ligand-independent activation of overexpressed but otherwise unaltered RTKs. We characterize overexpression as a “potentially oncogenic alteration” throughout the manuscript. However, experimental proof of that event is not easy to obtain in cultured cells and much more difficult to prove in tissue samples or animal models. Consistent with our long-term goals, we focused effort on experiments that would convey reasonable evidence of pathway activation and were most likely to enable the development of reliable assays (direct or proxy) for tissue samples.

Minor issues

1. It is unclear how the cell lines, RTKs, and ligands analyzed in Figure 4b were chosen, particularly given the apparent occasional incongruence between transcription data shown in Figure 4a and the protein expression data shown in Figure 4b.

Author Reply:

The nine BCa cell lines analyzed in Figure 4B are a subset of the 15 cell lines analyzed in Figure 4A; the number was reduced from 15 to 9 for practicality, but the subset includes each of the molecular phenotype classifications shown in Table 1. The subset is weighted toward the more aggressive phenotypes (basal squamous and neuronal) out of an interest in the potential coincidence of multiple ligand/RTK pair co-occurrences that were suggested by our analysis of the TCGA dataset [22] (Table 2A and Table S2A). Table S3A shows significant co-occurrence of group 1 TKs (i.e. TKs whose overexpression was associated with lower survival among the TCGA patients [22]), so the potential for multiple group 1 ligand/RTK pair co-occurrences in BCa cell lines was plausible.

Regarding the pathways chosen, as shown in Table 2A, the Gas6/AXL, CSF1/CSF1R and MST1/MST1R pathways have the highest Odds Ratio (Log2) values for co-occurrence, so these were prioritized in the immunoblot analysis in Figure 4B.

We regret that journal word limits that we encounter frequently (though not for PLOS ONE) have had a lasting negative impact on our tendency to provide detail. We have revised the Results text (p. 17, lines 465-469) to include our rationale for these choices.

2. Is there strong congruence between the cell line gene expression data reported in this work (Fig 3A and Fig 4A) and data available from the Broad CCLE? Similarly, if other biomarkers (from ref 21 cited in this work) characteristic of the N, BS, L, LI, and LP subtypes are applied to Broad CCLE data for the 15 BCa cell lines, are the subtype assignments the same as those reported in Table 1?

Author Reply:

We thank the reviewer for raising this interesting question. To address it, we downloaded the file “CCLE_expression.csv” from the DepMap portal (Source: Broad Institute; https://depmap.org/portal/download/). Per the DepMap web page, the file contains “RNAseq TPM gene expression data for just protein coding genes using RSEM. Data is Log2 transformed, using a pseudo-count of 1. Rows correspond to cell lines (Broad IDs) and columns correspond to genes (HGNC symbol and Entrez ID)”. Of the 15 BCa-derived cell lines used in our study, SLT3, FL3, T24T and T24M2 were absent from the CCLE dataset. Of the TKs analyzed in Fig. 3A, the CCLE dataset omits DDR1 and LYN. Using all other TKs in Fig. 3A, we composed the scatter plot shown below, with mRNA expression as determined by RT-PCR (Y.H. Lee et al., x-axis) vs. mRNA expression as determined by RNAseq (CCLE, y-axis). Displaying both axes using log2 scaling allows resolution of all data points. The data are reasonably concordant, considering the differences in method, date of analysis and established variability among the same cell lines cultured in different laboratories at different times:

[PLEASE SEE THE ATTACHED FILE "Response to Reviewers.pdf" to view the figure placed here]

Regarding whether the biomarkers used by Robertson et al. (our cited reference 22) characteristic of the N, BS, L, LI, and LP subtypes, if applied to Broad CCLE data for the 15 BCa cell lines, would result in the same subtype assignments as those reported in Table 1: while this is also an interesting question, it would take considerable time to address. Robertson used >3000 genes to generate their phenotypic groupings. Considering the work involved, we would be compelled to include all BCa cell lines for which data was available, an effort that we feel is beyond the scope of the current manuscript. Even if we were to consider it for an addendum, we would caution readers that real biological differences between cultured cell lines and tumor tissue samples affect the reliability and potential utility of this type of characterization, regardless of how definitive the approach might be.

We asked whether a limited set of differentially expressed TKs could be used to classify BCa cell lines into established molecular phenotypes and the results show that this can be done. The utility of this classification may be limited to TK-related studies, but in view of the importance of TKs as oncology targets and TK inhibitors as a drug class, it is still appreciable.

3. The authors’ attempts at brevity are greatly appreciated, particularly given the scope of this work. However, the paper appears to be a bit underwritten. Additional subheadings in the results section would improve readability. Likewise, additional transitioning between subsections and between and within paragraphs would improve clarity. For example, on page 14 (paragraph 1), the paper reads “cabozantinib inhibited 17 RTKs”. Yet, on page 14 (paragraph 2), the paper reads “All 19 ctRTKs displayed potentially oncogenic alterations”. It is not clear why 19 RTKs are being studied given that only 17 RTKs are inhibited by cabozantinib. Later, in paragraph 2 of page 14, the manuscript indicates that “patients harboring overexpression and/or amplification among 14 ctRTKs had significantly worse OS and DFS”. It is not apparent how these 14 RTKs were chosen.

Author Reply:

We appreciate the possible confusion here and we apologize for being overly brief. We have added text to the manuscript to provide greater detail regarding our goals and rationale and/or better clarify our findings (Introduction, pp. 4-5, lines 82-94; Materials and Methods, p. 6, lines 141-144; Results, p. 14, lines 370-374 and p. 17, lines 465-469).

Regarding the number of cabozantinib targets: 16 targets were known prior to our work and we identified DDR1, DDR2 and NTRK3 that had not been reported previously, bringing the total to 19. We have revised the Results text to clarify this point (pp. 15-16, lines 422-429).

The 14 RTK cabozantinib targets (ctRTKs) associated with significantly worse OS and DFS are group 1 and group 3 TKs only. Group 2 TKs are associated with improved survival relative to the other groups (Fig. 2B and Table S2C) and three group 2 TKs are also cabozantinib targets (DDR1, MERTK and MST1R), so the combined groups 1-3 ctRTK set is not associated with significantly worse outcome. We have revised the Results text to clarify this point (p. 16, lines 444-448 ).

4. Finally, the manuscript appears to occasionally use TK instead of RTK.

Author Reply:

This is intentional: RTKs are a subset of TKs, so only in TK sets that are exclusively RTK do we use that abbreviation. Since abbreviations used in the paper are listed separately in a footnote, we did not elaborate further on this in the text.

---

## [Decision Letter · Decision Letter 1]

10 Jun 2021

Autocrine Signaling by Receptor Tyrosine Kinases in Urothelial Carcinoma of the Bladder

PONE-D-20-33596R1

Dear Dr. Bottaro,

We’re pleased to inform you that your manuscript has been judged scientifically suitable for publication and will be formally accepted for publication once it meets all outstanding technical requirements.

Kind regards,

Pablo Garcia de Frutos

Academic Editor

PLOS ONE

Additional Editor Comments (optional):

Reviewers' comments:

Reviewer's Responses to Questions

**Comments to the Author**

1. If the authors have adequately addressed your comments raised in a previous round of review and you feel that this manuscript is now acceptable for publication, you may indicate that here to bypass the “Comments to the Author” section, enter your conflict of interest statement in the “Confidential to Editor” section, and submit your "Accept" recommendation.

Reviewer #1: All comments have been addressed

Reviewer #2: All comments have been addressed

Reviewer #3: (No Response)

2. Is the manuscript technically sound, and do the data support the conclusions?

Reviewer #1: Yes

Reviewer #2: (No Response)

Reviewer #3: Yes

3. Has the statistical analysis been performed appropriately and rigorously? 

Reviewer #1: Yes

Reviewer #2: (No Response)

Reviewer #3: Yes

4. Have the authors made all data underlying the findings in their manuscript fully available?

Reviewer #1: Yes

Reviewer #2: (No Response)

Reviewer #3: Yes

5. Is the manuscript presented in an intelligible fashion and written in standard English?

Reviewer #1: Yes

Reviewer #2: (No Response)

Reviewer #3: Yes

6. Review Comments to the Author

Reviewer #1: (No Response)

Reviewer #2: (No Response)

Reviewer #3: The attached file reiterates the answers to the questions above. The attached file also contains detailed comments for the authors.

7. PLOS authors have the option to publish the peer review history of their article (what does this mean?). If published, this will include your full peer review and any attached files.

Reviewer #1: **Yes: **Jesus M Paramio

Reviewer #2: No

Reviewer #3: No

---

## [Editor Report · Acceptance letter]

12 Jul 2021

PONE-D-20-33596R1 

Autocrine signaling by receptor tyrosine kinases
 in urothelial carcinoma of the bladder 

Dear Dr. Bottaro:

I'm pleased to inform you that your manuscript has been deemed suitable for publication in PLOS ONE. Congratulations! Your manuscript is now with our production department. 

Kind regards, 

on behalf of

Dr. Pablo Garcia de Frutos 

Academic Editor

PLOS ONE